# The influence of non-state-owned shareholder governance on the governance level of state-owned enterprises——Based on the perspective of board resolution behavior and party organization governance

Zhibin Zhang[1]*, Lishu Zhang[2], Aihua Xiong[3]

1 School of Economics, Shandong Technology and Business University Talent Introduction Project, Yantai, Shandong, China, 2 School of Economics, Shandong Technology and Business University, Yantai, Shandong, China, 3 School of Business Administration, Shandong University of Finance and Economics, Jinan, Shandong, China

* zhangzhibin2022@126.com

**Data Availability Statement:** All relevant data are within the Figshare website (https://figshare.com/) doi:10.6084/m9.figshare.25158251

## Abstract

With an increasing number of heterogeneous shareholders participating in corporate governance in reality, the assumption of shareholder homogeneity in agency theory is gradually relaxing in the modern field of corporate governance. The policy of mixed ownership reform in China provides empirical evidence for studying heterogeneous shareholder governance. To fully understand the governance effects of non-state shareholders, we employ the ownership proportion held by non-state shareholders among the top ten shareholders and the appointment of directors as measures for non-state shareholder governance. Using a panel fixed-effect model from the perspective of state-owned enterprises (SOEs) party organizations, we examine the impact of non-state shareholder governance on the governance level of SOEs. The study reveals that non-state shareholder governance positively affects the governance level of SOEs, with board resolutions playing a crucial role in this relationship. When party members serve as directors, the governance effect of non-state shareholders is more significant. Based on the aforementioned research findings, we recommend further refining corporate governance measures for SOEs within the context of SOE reforms. It is advisable to optimize the party organizational governance structure and leverage the synergistic effects of non-state shareholder governance and party organizational governance. Advancing reforms along the Pareto improvement path will contribute to establishing a distinctive corporate governance system for Chinese SOEs.

## Introduction

The heterogeneous ownership structure has gradually relaxed assumptions in existing corporate governance policies and practices based on traditional theories. With different resource

**Funding:** This work was supported by the Key Projects of Shandong Provincial Social Science Planning, Research Grant number 21BGLJ02, and Shandong Technology and Business University Talent Introduction Initiation Project, Research Grant number 306509. The funder (Zhibin Zhang) had role in formal analysis, data collection and methodology, design and data analysis, or writing-original draft of the manuscript.The funder (Aihua Xiong) had role in conceptualization and validation of the manuscript.

**Competing interests:** The authors have declared that no competing interests exist.

endowments and governance objectives, it ultimately affects a firm's financial behavior and operational performance [1]. Governance by non-state shareholders falls under the category of heterogeneous shareholder governance, and currently, there is limited theoretical research in the field of corporate governance regarding heterogeneous shareholder governance behavior [2]. For instance, Williams and Ryan (2007) found that greater disagreements among heterogeneous shareholders may pose challenges to executive performance evaluation and monitoring [3]. Additionally, the presence of heterogeneous shareholders makes it difficult to balance the diverse interests of company shareholders, leading management to prioritize the interests of shareholders whose preferences and risk tolerances align more closely with their own. Similarly, Goranova and Ryan (2022) emphasize that executives' vested interests do not necessarily aggregate and balance the interests of heterogeneous shareholders but rather prioritize the interests of those shareholders whose interests align more closely with their own [4]. Therefore, addressing issues related to heterogeneous shareholder governance is of utmost importance.

Given the agency conflicts arising from the "lack of owners" and "internal controllers" in SOEs [5], the reform of mixed ownership has become a vital aspect of the overall SOE reform in China. Drawing upon agency theory and resource theory, the participation of non-state shareholders in SOEs has a significant impact on these enterprises. Scholars have pointed out that non-state shareholder governance can alleviate policy burdens [6], enhance economic benefits [7], facilitate corporate innovation [8], and promote social responsibility [9]. However, due to government intervention resulting from the multiple objectives of SOEs [10], practical measures have often prevented capable and influential non-state shareholders from effectively participating in SOE management [11]. In January 2021, Peng Huagang emphasized the importance of promoting SOE reform through mixed ownership, leveraging the positive effects of non-state shareholders, and enhancing the governance level of SOEs. To curb administrative governance in SOEs and improve their governance capacity, this study argues that it is necessary to examine the impact of non-state shareholder governance on SOEs.

As an important decision-making body within a corporate enterprise, the board of directors serves as the core of corporate governance. In the current context of mixed ownership reform, an increasing number of non-state shareholders are appointing directors, supervisors, managers, and other senior executives to SOEs, enabling them to actively participate in corporate governance decision-making, operational management, and internal oversight. Directors, acting as representatives appointed by shareholders, are expected to vote in line with the shareholders' opinions, while overseeing management and safeguarding shareholders' interests remain their primary responsibilities [12]. However, the synergistic effects of mixed ownership reform and party organization governance, which are institutional innovations aimed at promoting the high-quality development of SOEs, have not been thoroughly analyzed thus far. We aim to address this issue by examining the level of board governance.

Compared with SOEs, non-state shareholders have stronger motivation to participate in corporate governance based on the pursuit of profit maximization. However, Zheng (2021) pointed out that the practical measures launched by the mixed ownership reform of SOE did not make capable and powerful non-state shareholders truly participate in corporate governance [11]. Zheng (2021) theoretically discussed how to make non-state-owned shareholders incentive-compatible and willing to participate in the mixed-ownership reform of SOEs through the design of corporate governance systems. He analyzed three aspects: appointing directors to participate in corporate governance, forming a limited partnership structure with private capital background strategic investment, and transforming state-owned assets into preferred shares. We differ from Zheng (2021) in the followings ways. First, Zheng (2021) focus on incentive compatibility of non-state shareholders. However, our focus is on the governance

effects of non-state shareholders and their governance relationship with party organizations. Second, we not only analyzed it theoretically, but also tested it empirically [11]. In the face of various contradictions, whether there is an effective governance method for non-state shareholders to participate in the governance of SOE is an important problem currently faced by the academic and practical circles. At the same time, in 2016, General Secretary Xi put forward the principle of "two consistency" ("two consistency", that is, "Adhering to the party's leadership over SOEs is a major political principle that must be followed consistently; establishing a modern enterprise system is the direction of SOE reform and must be followed consistently."), which integrates the party's leadership into all aspects of corporate governance, and embeds the corporate party organization into the corporate governance structure. However, the functions, powers and responsibilities of party organizations participating in corporate governance are not clear. Its "discussion priority" system has the phenomenon of "preempting" the political, social and economic goals of state-owned enterprises. It limits the control of the board of directors over corporate operations, thereby inhibiting Participation in board resolutions. It can be seen that the reform of SOE is a systematic project, which not only requires non-state shareholders to actively participate in corporate governance, but also needs to optimize the party organization governance system. In 2015, the Central Committee of the Communist Party of China (CPC) and the State Council jointly issued the "Guiding Opinions on Deepening the Reform of State-Owned Enterprises." This document outlined a framework for defining functions, categorizing, and progressively implementing reforms in SOEs. The suggested approach involved categorizing SOEs into competitive and non-competitive types, with non-competitive entities further classified as natural monopolies or serving public welfare goals. Non-competitive SOEs primarily align with national policy objectives. Competitive SOEs, on the other hand, prioritize market profitability goals. Considering relevant theories in traditional corporate governance, our research focuses on competitive state-owned enterprises. Based on this, how to regulate the governance between the non-state shareholders of SOE and party organizations to maximize the advantages of mixed ownership reform? This article will explore these issues based on the board governance level.

We use a large sample of China-listed SOEs in the manufacturing industry for 2009–2019, extracted from China Center for Economic Research (CCER) database, CSMAR database and annual reports of listed companies. We empirically analyze the governance effect of non-state shareholders based on board resolution and the moderating role of party organizations in SOE.

We begin our analysis with a baseline model in Ordinary Least Square (OLS). We find that the greater the shareholding of non-state shareholders at the level of ownership structure, the higher the governance level of SOE. We have also tested that the results are not subject to specific modeling choices regarding sample selection, outcome construction, and standard error adjustments. The baseline model with the shareholding structure variable is the extended to account for more flexible treatment intensity measures along high-level governance dimensions. In 2019, the "Operation Guidelines for Mixed Ownership Reform of Central Enterprises" promulgated by the State-owned Assets Supervision and Administration Commission stipulated that "as far as possible, non-state shareholders can send directors or supervisors", which changed the traditional concept that SOE "have the final say" by the government from the policy level.

These findings remain consistent following the application of a large number of robustness tests, including: 1. Alternative measure of independent variable. 2. Replace the regression model. 3. Change time sample. We then employ two additional tests to overcome endogeneity concerns. First, we employ the instrumental variable approach. Second, we conduct a Propensity Score Matching (PSM) analysis to address the selection bias. In all tests, we find consistent results with our baseline findings. Having established a positive association between non-state

shareholder governance and governance level, we analyze the mediating effect of board resolutions. We find that non-state shareholder governance can improve SOE governance through board resolutions. We further analyze the moderating role of party organization governance. Our research results show that when members of the party organization serve as directors, the governance of non-state shareholder is more significant effect on the governance level of SOE.

Consequently, the current paper seeks to make the following contributions to the existing literature. Firstly, the study deepens the understanding of mixed ownership reform, revealing its mechanisms and effects in enhancing the governance efficiency of SOEs. We add to the literature documenting the economic benefits of non-state shareholder governance in SOEs. Several recent studies have examined the impacts of non-state shareholder governance on innovative decision-making [13], operating performance [14], and technology innovation [15]. Research on non-state shareholder governance and governance level of SOE, what is not been thoroughly examined in detail thus far are the reasons for such outcomes. In addition, although there are many literatures on the level of corporate governance, most of them study corporate governance as a pre-variable [16–18]. Taking Zhejiang enterprises as the research object, Du et al. (2022) analyzed the governance effect evaluation of the mixed ownership reform, but there is insufficient research on the improvement path of its governance capacity [19]. We take corporate governance level as a post-variable It is very important in SOE to explore the impact of non-state shareholder governance on governance level. We complement these studies by providing evidence that non-state shareholder governance promotes governance level of SOE.

Further, we identify the mechanisms of non-state shareholders on governance level from the perspective of board resolutions. One of the most critical aspects of our results is that, in addition to highlighting the influence of non-state shareholder governance on governance level, we also represent indirect evidence of the influence of non-state shareholder governance and board resolutions on governance level of SOE. Thus, our results highlight how non-state shareholder governance interacts with board resolutions and the governance level.

Secondly, by means of policy measures, the study provides empirical support for heterogeneous shareholder governance, bridging the gap between theory and practice. We place party organization governance within the framework of non-state shareholder governance and SOE governance. Party organization and mixed ownership reform are unique systems of Chinese SOE, and most of the relevant literature studies them separately. As important stakeholders, whether important members of the party organization and non-state shareholders/directors can exert the governance effect of "1+1>2" remains to be tested. Therefore, combined with the mixed ownership reform system, we incorporate the party organization into the analytical framework of corporate governance at the micro level of SOE, explore the relationship between non-state shareholders and party organization governance, and provide theoretical and empirical support for it.

Thirdly, it highlights the significance of governance boundaries, emphasizing the correlation between party organization governance and governance by non-state shareholders. In terms of direct effects, our evidence shows that different shareholders or directors from different types of companies can improve the governance level of SOE and be more active in the board resolution process. After introducing the variable of party organization governance, we found that when there are important members of the party organization on the board of directors, the positive impact of non-state shareholder governance on the governance level of SOE is suppressed. If SOEs want to further improve the level of governance, one strategy may be to clarify the boundaries of party organizations' governance rights, responsibilities, and interests, to make them legal, and to jointly exert the governance effects of mixed ownership reform and party organizations.

Lastly, the study puts forth concrete reform recommendations, offering targeted guidance for mixed ownership reform and promoting the enhancement of governance levels in SOEs. In summary, this research holds profound significance in advancing theoretical understanding, providing empirical support, and offering specific recommendations for practical implementation.

## Literature review

### Research on the non-state shareholder governance

A large literature studies the role and influence of non-state shareholder governance [for an overview [20]]. In China, on-state shareholder governance can effectively improve the "lack of ownership" and "insider control" of SOEs, integrate resources and optimize governance mechanisms, and improve the business development capacity of state-owned enterprises. Much of the literature focuses on the role of non-state shareholders in the shareholding structure and board structure. The shareholding structure dimension refers to the proportion of equity held by non-state shareholders in SOEs. The board structure level refers to the appointment of persons by non-state shareholders as directors of SOEs. Several studies have analyzed how non-state shareholder governance influence Cash holding levels [7], overinvestment [21], executive corruption [15]. Hao and Wang (2015) argue that non-state shareholders can both vote on major decisions at the shareholders' meetings of SOEs and effectively participate in the management governance of SOEs by nominating [22], electing and appointing directors to SOEs. As a result, more scholars believe that the governance effect is better served by non-state shareholders appointing people as directors than at the level of shareholding structure [e.g., for the pay performance sensitivity [23]; for the internal quality control [24]; for the transparency of accounting information [25]; for the technological innovation [26]]. From these studies picture emerges that non-state shareholder governance is beneficial to SOEs, both at the level of shareholding structure and board structure.

Recent years witnessed a surge in heterogeneous shareholder governance of firms on a world-wide scale. Goranova and Ryan (2006) argues that participation of heterogeneous shareholders is a preferred institutional model for the development of firms towards good discovery [4]. Under China's SOE reform system, more and more heterogeneous shareholders are involved in the management and governance of SOEs [27–29]. Studies find that firms supervised or managed by shareholders or boards with heterogeneous backgrounds tend to the good governance characteristics [30]. They are fewer agency costs [31], are more technological innovations [32], and are more efficient with investments [33]. Among the reasons mentioned is that heterogeneous shareholders or directors improve corporate governance and management practice. This phenomenon is because the heterogeneous shareholder or director mechanism has the characteristics of flexibility, independent decision-making, market sensitivity, cost control, and perfect management. Building on this evidence, we examine the impact of non-state shareholder governance on the governance level of SOEs.

### Research on the party organization

In recent years, in the field of corporate governance, more and more scholars have studied the issues related to the governance of party organizations, and the governance of party organizations has had a certain impact on the corporate governance of enterprises. At the level of corporate governance, the governance of party organizations has the role of supervision and checks and balances. On the one hand, party organization governance can improve the level of corporate governance and the efficiency of the board of directors [34], thereby reducing agency costs [35], restraining the hollowing out behavior of major shareholders, and

preventing the loss of state-owned assets [36], so that the interests of the company are not infringed [37]; on the other hand, the governance of the party organization can also affect the decision-making behavior of enterprises [38], and strengthen the high-quality audit of enterprises to make corporate governance more self-disciplined [39, 40]. Due to the importance of the party organization in the company's organization, the development goal of the company has gradually changed from a single profit maximization orientation to a dual orientation of economic benefits and social value, and internalized social and political goals into the company's strategic decision-making and daily operations. In action, Liu et al. (2022) believed that the governance of party organizations has improved the level of ESG investment of enterprises, and assumed important responsibilities in non-market aspects such as the environment and society [6]; at the same time, Xiu et al. (2022) policies to effectively promote state-owned enterprises to implement poverty alleviation strategies [41]. In the context of mixed ownership reform, Wang et al. (2019) pointed out from the perspective of state-owned enterprise equity that there is an interactive effect between party organization governance and state-owned equity changes, and that party organization governance can affect the effect of state-owned equity changes on corporate governance [42].

What is of interest is that this paper finds that more scholars have provided a more mature theoretical and empirical analytical framework for exploring the governance of non-state shareholders and party organisation governance. Based on the existing literature, it is not difficult to find that the entry of non-state shareholders into SOEs exerts governance effects and monitoring effects. While there is a continuing body of literature on the governance and economic effects of non-state shareholders on SOEs, little research has been conducted on the impact of non-state shareholder governance on board resolution and the board governance effects of non-state shareholders based on party organisation governance, which merits further study.

## Theoretical analysis and research hypothesis

**Shareholding structure and corporate governance level.** Due to historical reasons, Chinese SOEs have long faced the issue of "absentee ownership." There is a lack of effective supervision and management within the enterprises, which makes them susceptible to the phenomenon of "insider control". Based on the principal-agent theory, the ultimate control of SOE belongs to government departments, there are many political goals and planning mechanisms. Executives are usually appointed by higher-level government departments, with the administrative level of government officials, and have a strong administrative color. Meanwhile, in Chinese governance system, central or local governments appoint and evaluate government officials, while administrative and economic affairs, including those of SOEs. Based on a tournament-style promotion mechanism, senior executives are not overly concerned about affecting corporate governance.

Under the mixed-ownership reform system, SOEs in China have introduced a large number of non-state shareholders. The positive governance effects of non-state shareholders have been confirmed through numerous investigations [7]. The supervision of non-state shareholders restrains managers' irrational investment motives and reduces agency costs [13, 33]. Secondly, the external environment holds a positive attitude towards the governance of non-state shareholders. Research conducted by Beck (2022) has found that the introduction of non-state capital in state-owned enterprises can convey positive signals in the capital market [43]. This indicates that non-state shareholders actively participate in corporate governance and generate positive market reactions.

Stakeholder theory suggests that the management activities of a company's business managers balance the interest requirements of each stakeholder in a comprehensive manner [44]. In

the internal environment of enterprises, the participation of heterogeneous equity subjects in corporate decision-making can expand the set of opportunities for decision-making. Moreover, it also contributes to the specialization of decision-making and capital and risk-taking, which is conducive to forming a democratic and scientific decision-making mechanism and improving corporate risk control [30]. As Bennedsen and Wolfenzon (2001) note [45], when there are multiple heterogeneous shareholders in a company, they can make the right business decisions and strengthen corporate governance capabilities.

**Board structure and corporate governance level.** Non-state shareholders participate in SOE through mixed ownership reform, and the ownership structure has undergone substantial changes, and then assign personnel to participate in the corporate governance, and truly enter the internal decision-making level of SOE. According to the upper echelon theory, the heterogeneity of top members leads to divergent opinions, the greater the likelihood that corporate funds will be spent in the right direction [45]. In SOE, non-state shareholders participate in SOE, and even appoint personnel to the board of directors, which can directly supervise the management and improve corporate governance capabilities. Assume that the company's relevant actions do not comply with corporate governance procedures. Non-state shareholders or directors can express their opinions and amend the content of the proposal. The governance behavior of non-state shareholders alleviates the agency costs within state-owned enterprises question. Based on the principle of profit maximization and businessmen's "pursuit of interests" rule, the governance behavior of non-state shareholders "voting with their hands" helps to achieve the role of supervision and checks and balances [46], thereby improving the governance level of SOE.

Based on upper echelons theory, managerial characteristics influence strategic choices of companies [47]. Managers with different background traits will have different values and personal perceptions. These factors will directly influence their communication and cooperation at work, indirectly influencing the related decisions and, consequently, the company's behavior. Meanwhile, resource dependence theory notes that board composition is essential to the board's ability to provide governance to the company [20]. As Pfeffer (1972) considers [48], the composition of the board of directors is not random but a logical response to the company's internal environment.

Based on the above discussion, we state our central hypothesis as follows:

**H1A.** There is a positive association between non-state shareholder and governance level of SOE. That is, the non-state shareholder can improve governance level of SOE.

**H1B.** There is a positive association between non-state director and governance level of SOE. That is, the non-state director can improve governance level of SOE.

Board resolution is an important way for the board of directors to exert its governance effect, and the voting of the board of directors is the most direct evidence to reflect the board resolution. In this process, board resolution has become an organic unity of directors' participation in decision-making and checks and balances. Due to the "pyramid" hierarchical relationship of SOE, if non-state shareholders have no "speaking power" in the process of operating and decision-making, they will evolve into the decision-making of upper-level SOE, resulting in high time cost of decision-making matters and greatly reducing decision-making efficiency. Even lead to the loss of market opportunities for enterprises. The reasons for directors appointed by non-state shareholders to participate in board resolutions are mainly reflected in two aspects: First, in the face of the complex and volatile internal and external environment of enterprise, managers from non-state enterprise can enter the board of directors of SOE in a more professional and market-oriented manner. Participate in corporate governance decision-making with an attitude of participation in corporate governance decisions, so as to make multi-faceted evaluations of corporate decision-making projects, and give full play to the advantages of board

resolutions [23], rather than voting resolutions only as "formalism"; Second, non-state share-holders appoint persons to serve as directors of SOE. The voting behavior of non-state directors in board resolutions can also affect the "insider control" of SOE, forming a certain degree of checks and balances, which is conducive to alleviating the moral hazard of "semi-market-ori-ented, semi-administrative" management of SOE [49]. Based on the upper echelons theory, non-state shareholders can give full play to the functions of the board of directors by appointing directors to supervise the internal personnel of SOE. It highlights the function of the board of directors to correct unfair decisions in a timely manner in the process of corporate governance, and finally realizes the modernization of the governance level of SOEs.

In order to play a role in the governance process of SOE, non-state shareholders not only participate in SOE, but also appoint personnel to send directors to SOEs, and supervise and balance the controlling shareholders and management through the board of directors to safe-guard their own interests [12]. According to research, it is found that scientific and effective board resolutions need to be discussed by directors from different backgrounds, because direc-tors from different backgrounds can represent different stakeholders and can think about issues from different perspectives, which is more conducive to making resolutions more rea-sonable and effective [50]; based on the high-level ladder theory, the heterogeneous back-ground of board members can lead to different opinions [47]. The presence of a non-state shareholder on the board or a dissenting vote by a director implies greater diversity of opinion. For example, in the third board resolution in 2019, "Yunnan Baiyao" director Song Jian voted against the proposal for major asset restructuring that did not consider shareholders' interests. Compared with SOEs, managers of non-state-owned enterprises often have market-oriented and professional judgments. Non-state shareholder governance can effectively supervise and modify the decision-making behavior and management activities of SOEs, and promote the rationalization of board resolutions and proposals. Therefore, we make the following assumption:

**H2.** The governance of non-state shareholders can improve the governance level of SOE through the board of directors' resolution behavior, that is, the board of directors' resolution behavior exerts an "intermediary effect".

Among state-owned enterprises, the company's management and governance place special emphasis on the leadership of the party. In October 2016, General Secretary Xi Jinping made a clear request in his speech at the national state-owned enterprise party building work confer-ence to improve the leadership system of "two-way entry and cross appointment". "two-way entry, cross-serving", mainly means that the secretary of the party committee (party group) and the chairman are held by one person, the members of the party committee respectively enter the board of directors, the board of supervisors and the management team through legal procedures, and the party members of the board of directors, the board of supervisors and the management team enter the party committee in accordance with relevant regulations meeting. It is necessary to give full play to the "leading role" of the party organization and assume the responsibility of "setting the direction, managing the overall situation, and ensuring imple-mentation". In June 2015, the "Regulations on the Work of the Party Group of the Communist Party of China (for Trial Implementation)" promulgated by the Central Committee of the Communist Party of China first proposed the "pre-discussion" mechanism of state-owned enterprise party committees; The "Notice on the Spiritual Key Tasks of the Construction Work Conference" officially established "discussion beforehand" as a decision-making mechanism that must be adopted by all SOE. The party organization governance of "pre-decision-making" behavior is usually regarded as an alternative mechanism to control and supervise the marketi-zation process of SOE [42]. It participates in major decision-making of the company's manage-ment and governance before the resolution of the board of directors, supervises the

implementation of the policies and guidelines of the party and the state within the enterprise, and the party organization is endowed with major powers. The matters under "discussion before" involve a series of issues such as company budget and final accounts, corporate asset reorganization, transfer of property rights, capital operation and large-scale investment, which to a certain extent lead to the overlap between the matters resolved by the board of directors and the matters discussed by the party organization. This procedure necessarily inhibits the participation of directors in board resolutions. At the same time, in the process of party organization governance, "two-way entry, cross-serving" is the main way for the party organization to intervene in the governance of the board of directors, especially when the party secretary or deputy secretary of the party committee is "cross-serving" in the board of directors, it is given a greater decision-making power. Based on the Confucian culture under traditional Chinese thought, ethics and harmony are deeply rooted in each individual organization; at the same time, data shows that more than 50% of SOE party secretary and chairman overlap. Based on traditional culture and realistic background, party organization governance will inevitably inhibit the participation of non-state-owned directors in the board of directors decision-making process. Therefore, we propose the following assumption:

**H3.** Compared with the existence of important members of the party organization on the board of directors, when the board of directors has no important members of the party organization, the governance of non-state-owned shareholders can improve the governance level of SOEs more significantly.

In the process of corporate governance, "principled dissent" will promote the development of enterprises [51]. The directors express their opinions on the resolutions of the board of directors, which is conducive to realizing the supervision of the state-controlled shareholders and alleviating the problem of information asymmetry. Non-controlling shareholder directors participate in the decision-making process of the board of directors by voting against or abstaining from voting, realizing information supervision and the role of information media. On the one hand, this is not only conducive to the supervision of information generation, but also conducive to the transparency of information transmission, thus forming a certain degree of checks and balances and constraints on the internal personnel of state-owned enterprises. On the other hand, this helps to restrain the earnings management activities of "semi-official" executives with equal motives for political purposes, thereby alleviating the information asymmetry among various entities and ultimately optimizing the governance level of state-owned enterprises. If the resolutions of the board of directors include matters that may harm the interests of shareholders and enterprises, directors appointed by non-state shareholders not only have a strong incentive to supervise, but also stimulate other directors of non-controlling shareholders to express their own opinions. Opinions, vote non-yes to influence or even reject the proposal, which helps to improve the content of the proposal [52], and inhibits the inefficient investment and in-service of SOE insiders based on the motivation of political tournaments and building business empires. Short-sighted behaviors such as consumption are not conducive to the realization of high-quality development of SOE, thereby improving the quality of internal governance. It is easy for SOE to form scientific and reasonable boards of directors through mixed ownership reform, and directors representing different stakeholders can give full play to their own opinions when making resolutions on board meetings, which makes the discussion and decision-making more reasonable and effective [53]. Gomes and Novaes (2005) believe that rational directors with non-state background can participate in board resolutions through "voting by hand" [50], and through market-oriented and professional perspectives, they can bring differentiated and reasonable suggestions to board resolutions, effectively supervise and The operation and management activities and decision-making behaviors of SOE are revised, thereby improving the effectiveness of SOE's decision-making, and finally

establishing the status of independent market players. Based on this, this paper proposes the following assumption:

**H4.** During the governance process of non-state shareholders, the decision-making behavior of the board of directors has a significant positive correlation with the governance level of SOE.

In summary, based on stakeholder theory and upper echelon theory, we theoretically analyze how non-state shareholder governance influences the governance level of SOEs through the ownership structure and the board structure. Subsequently, we examine the potential channel of non-state shareholder governance on the governance level of SOEs from the perspective of board resolution behaviors. Finally, we analyze the differential impacts between non-state shareholder governance and governance level under party organization governance. The theoretical framework is illustrated in Fig 1.

## Data sources and description

### The sample

We first obtain financial data for all manufacturing SOEs in competitive fields listed on the Shanghai Stock Exchange and Shenzhen Stock Exchange from 2009 to 2019 (Taking into account that non-competitive SOEs are found in industries such as mining, electricity, gas, water production and supply, construction, transportation, warehousing, and postal services, our research primarily focuses on the manufacturing sector). Given the problem of data selection bias, and In order to thoroughly examine the governance of heterogeneous shareholders in SOEs, listed companies are studied with SOEs listed before 2009, and those listed midway are excluded. We test the above predictions that SOEs are introduced to non-state shareholders at different times and varying degrees from 2010 to 2019. We started in 2009 to make sure the accounting information is comparable across years because the share-trading reform of

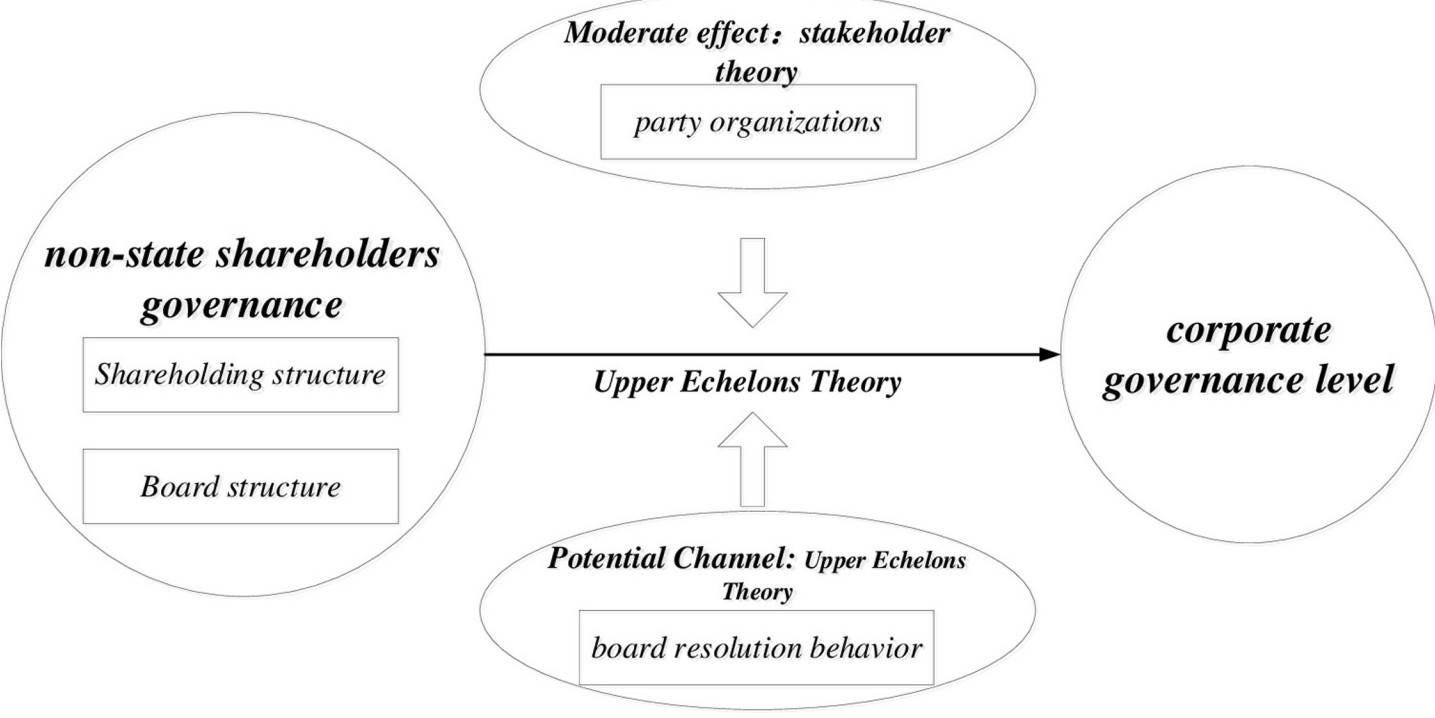

**Fig 1.**

public companies was completed in 2007, and the financial crisis happened in 2008. Following prior literature, we apply several criteria to screen the sample: First, we delete observations under special treatment by stock exchanges (labeled ST/*ST/PT). Second, we eliminate missing or outlier observations. Finally, the continuous variables of the sample data are winsorized at 1% above and below. After these steps, our final sample consists of 4101 firm-year observations.

The heterogeneous governance shareholders are divided into ownership and board structure levels. Specifically, ownership structure data are taken from CCER China Economic and Financial Database, and board structures are hand-collected from their companies' board resolutions reports. Other financial and corporate governance data are retrieved from the China Stock Market & Accounting Research (CSMAR) database.

## Variable selection

**Governance level of SOE.**   To measure the effectiveness of corporate governance, governance level needs to be measured using multiple dimensions. Therefore, drawing on the study by Zhou et al. (2020), we applied principal component analysis (PCA) to construct the governance level from three dimensions: supervision, incentives, and decision-making. The proportion of independent directors and board size represent the supervisory role of the board, while the proportion of institutional ownership and the ownership balance (the sum of the shareholdings of the second to fifth largest shareholders divided by the shareholding of the controlling shareholder) indicate the supervisory role of the ownership structure. Executive compensation and the proportion of executive shareholdings represent the incentive mechanisms in corporate governance. The combination of the chairman and CEO positions indicates the decision-making power of the CEO. Based on these seven indicators, we utilized PCA to construct a corporate governance index (*gover*). The first principal component derived from PCA is used as a comprehensive indicator reflecting the level of corporate governance. It is important to note that PCA standardizes each component indicator, and the relative magnitude of the corporate governance index is more significant than its absolute value. A higher index value indicates a better corporate governance level.

**Non-state shareholder governance.**   The independent variable is non-state shareholder governance. Referring to [15], we classify shareholders into six sub-categories: 1. State-owned shareholders mainly refer to shareholders whose shares are directly held by government departments such as the Ministry of Finance and the State-owned Assets Supervision and Administration Commission or indirectly held by the state through holding companies and investment management companies; 2. Private shareholders mainly include the shareholders of private enterprises (non-financial industry) for investment; 3. Natural shareholders, specifically Chinese natural persons in China and foreign natural persons or families holding shares in the company formed by shareholders; 4.Foreign shareholders, including foreign natural persons and foreign legal persons (including China's Hong Kong, Macao, and Taiwan) to invest in the company's shareholders; 5. Institutional investors, mainly legal institutions engaged in securities investment (including insurance companies, pension funds, investment funds, securities companies, banks, etc.) 6. In addition to the above categories, other shareholders mainly include shareholders of universities, research institutes, or institutions that make investments in the company. We aggregate 2–6 for non-state shareholders. In the main effect test, we use ownership structure to measure the governance of non-state shareholders. The shareholding structure represents the degree of governance of non-state shareholders by the shareholding ratio (*nsta*) of non-state shareholders in the top ten shareholders. In the robustness test, the board structure (*pai_nsta*) is the natural logarithm of the number of board

members appointed by non-state shareholders plus 1 (the criterion for appointing board member was as follows. If a natural shareholder serves as a board member in a public company, it is deemed that the person appoints a senior executive to the company. Moreover, if board member of a listed company serves in the unit of a legal person shareholder, it is deemed that the legal person appoints a board member to the company).

**Board resolution behavior.**   According to the regulations of the Shanghai and Shenzhen stock exchanges, in the decision-making process of the board of directors of a listed company, directors with voting rights need to vote on relevant proposals. There are four voting options: agree, oppose and abstain. For negative votes, abstentions or avoidance votes, reasons must be given. Liu et al. (2020) believed that the directors' "dissenting votes" and "abstentions" were both dissenting results when studying the resolutions of the board of directors [54]. Gao (2021) showed that although directors abstained from voting and chose to vote for or against [55], most of the time abstentions played the role of "negative votes". Therefore, based on previous research, we set a dummy variable (*dissent*). We believe that both disapproval and abstention of board members are considered to be directors' participation in board resolutions. Directors' disapproval reflects strong opinions, while abstention is an expression of "silent resistance". Both will ultimately affect Result of board resolution. If the state-owned listed company has a non-favorable vote on the resolution of the annual board meeting or (interim) board meeting during the whole year, the value will be 1, otherwise, it will be 0.

**Party organization governance.**   If the party secretary or deputy secretary serves as a member of the board of directors (chairman, vice chairman or director), define *party* as 1, otherwise 0.

**Control variables.**   According to the existing research results, the control variables in this paper mainly include the control variables at the board level and the firm level. The control variables are as follows: (1) company size (*size*), (2) management's shareholding (*manage*), (3) asset-liability ratio (*lev*), (4) return on equity (*roa*), (5) net cash flow (*cfo*), (6) business growth (*grow*), (7) capital expenditures (*expend*). Detailed definitions of variables are presented in Table 1.

## Empirical model

To examine the effect non-state shareholder governance on governance level of SOE, we estimate the following regression model:

$$gover_{it} = \alpha_0 + \alpha_1 nsta_{it}/pai\_nsta_{it} + \alpha_Z Controls_{it} + year + \mu_i + \varepsilon_{it} \tag{1}$$

Using a panel fixed effects model. Where *i* and *t* refer to firm and year, respectively. The dependent variable is the governance level (*gover*). The independent variable is the non-state shareholder governance (*nsta/pai_nsta*). Control variables consist of the firm size (*size*), management's shareholding (*manage*), assets and liabilities (*lev*), return on assets ratio (*roa*), net cash flow (*cfo*), business growth (*grow*), capital expenditures (*expend*). Additionally, we control firm fixed effects ($\mu_i$), year fixed effects (*year*), The coefficient of interest is $\alpha_1$. We expect the coefficient to be positive, implying that non-state shareholder governance can help improve the governance level of SOEs.

## Descriptive statistics

In order to get a preliminary understanding of the basic statistical properties of the sample observations corresponding to each variable, Table 2 shows the descriptive statistical results of the relevant variables. The mean value of the governance level (*gover*) is -0.69, indicating that the governance level of SOE is low; the minimum value is -2.72, and the maximum value is

**Table 1. Variable definitions.**

| Variable | Symbol | Definition |
|---|---|---|
| Governance level of SOE | *gover* | Based on the research of Zhou et al. (2020) [56], we use the principal component analysis method to construct comprehensive indicators to measure the governance level of state-owned enterprises from three aspects (7 variables): supervision (proportion of independent directors, size of the board of directors, institutional shareholding ratio and equity balance), incentive (proportion of executive compensation and executive shareholding), and decision-making (whether the chairman and the general manager have two functions in one) |
| Non-state shareholder governance | *nsta* | Ratio of non-state shareholders among the top ten shareholders |
| | *pai_nsta* | Natural logarithm of one plus the number of board members appointed by non-state shareholders |
| Board resolution behavior | *dissent* | If the SOE has board objections in the annual board resolution announcement, it will be 1, otherwise it will be 0 |
| Party organization governance | *party* | If the secretary or deputy secretary of the party committee serves as a member of the board of directors (chairman, vice chairman or director), it is defined as 1; otherwise, it is 0. |
| firm size | *size* | Natural logarithm of the book value of total assets at the end of the year |
| management's shareholding | *manage* | Number of shares held by management/total share capital of the company |
| assets and liabilities | *lev* | The ratio of total liabilities to total assets |
| return on assets ratio | *roa* | Ratio of net cash flows from operating activities to total assets |
| net cash flow | *cfo* | Net cash flow from operating activities/total assets |
| business growth | *grow* | operating income growth rate |
| capital expenditures | *expend* | Capital Expenditure/Total Assets |

2.53, indicating that there are large differences in the governance level of different SOE. The average shareholding ratio (*nsta*) of non-state shareholders in SOE is 27%, the minimum value is 0, and the maximum value is 1, reflecting the mixed state-owned enterprises. Taking into account the "Decision of the Central Committee of the Communist Party of China on Major Issues Concerning Comprehensively Deepening Reforms" issued during the Third Plenary Session of the 18th Central Committee in 2013, which proposed "actively developing a mixed-ownership economy," there has been a significant transformation in the ownership structure of SOEs listed before 2009. When nsta = 1, it means that non-state capital has been introduced, accounting for 100% of the ownership in SOEs. The degree of ownership reform varies, and

**Table 2. Descriptive statistics.**

| Variable | N | Mean | Std. | P25 | P50 | P75 | Min. | Max. |
|---|---|---|---|---|---|---|---|---|
| *gover* | 4101 | -0.69 | 0.61 | -1.07 | -0.76 | -0.35 | -2.72 | 2.53 |
| *dissent* | 4101 | 0.04 | 0.18 | 0.00 | 0.00 | 0.00 | 0.00 | 1.00 |
| *party* | 4101 | 0.10 | 0.41 | 0.00 | 0.00 | 0.00 | 0.00 | 1.00 |
| *nsta* | 4101 | 0.27 | 0.25 | 0.09 | 0.17 | 0.35 | 0 | 1 |
| *pai_nsta* | 4101 | 0.62 | 0.55 | 0.00 | 0.69 | 0.69 | 0. | 2.71 |
| *size* | 4101 | 22.55 | 1.31 | 21.63 | 22.38 | 23.40 | 18.27 | 27.47 |
| *manage* | 4101 | 0.01 | 0.03 | 0.00 | 0.00 | 0.00 | 0.00 | 0.56 |
| *lev* | 4101 | 0.48 | 0.19 | 0.340 | 0.49 | 0.63 | 0.02 | 1.11 |
| *roa* | 4101 | 0.04 | 0.05 | 0.01 | 0.03 | 0.06 | -0.59 | 0.38 |
| *cfo* | 4101 | 0.05 | 0.07 | 0.01 | 0.05 | 0.09 | -0.47 | 0.48 |
| *grow* | 4101 | 0.37 | 2.25 | -0.05 | 0.11 | 0.35 | -1.55 | 67.49 |
| *expend* | 4101 | 0.05 | 0.04 | 0.02 | 0.03 | 0.06 | 0 | 0.55 |

some SOEs are privatized; the mean value of non-state shareholders appointed to senior management (*pai_nsta*) is 0.62, the minimum value is 0, and the maximum value is 2.71. The mean value of board resolutions (*dissent*) is 0.04, indicating that 4% of state-owned enterprises have dissent from the board of directors, that is, fewer state-owned enterprises vote against or abstain from voting on the board of directors. This is basically consistent with the research results of scholars such as Liu et al. (2020). The mean value of the party organization variable (*party*) is 0, and the 75th percentile is also 0, which means that most important members of the party organization of SOEs do not hold positions on the board of directors.

Judging from the control variables, the standard deviations of the size and growth of SOEs are 1.31 and 2.25. The total assets and operating income growth rates of the sample Chinese SOEs are quite different, which means that in the manufacturing industry, the scale of SOE is different, and the long-term development plans of enterprises are also different. There are both sunset industrial enterprises and sunrise industrial enterprises; management shareholding ratio (*manage*), asset-liability ratio (*lev*), corporate profitability (*roa*), the net cash flow (*cfo*), and the standard deviation of capital expenditure (*expend*) are small, and among the SOE samples differences are also lower. The descriptive statistics of the remaining control variables are generally consistent with existing studies [56, 57].

The correlation coefficient between the governance level of SOE is shown in Table 3, of which the lower left is the Pearson correlation coefficient, and the upper right is the Spearman correlation coefficient. The Spearman correlation coefficient and Pearson correlation coefficient of non-state shareholder governance and governance level of SOE (*gover*) at the level of ownership structure (*nsta*) and board structure (*pai_nsta*) are both significantly positive at the 1% level. The coefficient does not exceed the critical value of 0.6, and there is no multicollinearity. The hypothesis H1 is preliminarily verified. The governance of non-state shareholders improves the governance level of SOE. More convincing research conclusions need to be further tested by empirical analysis. The correlation coefficient between ownership structure (*nsta*) and board structure (*pai_nsta*) is as high as 0.6, and there is serious collinearity. The SOE reform policy mentions that "as much as possible, non-state shareholders can appoint directors, and actively promote the corporate governance of SOE." Therefore, board structure fits well as a proxy for ownership structure.

## Empirical results

### Baseline regression results

To test the impact of non-state shareholder governance on governance level of SOE, we use the shareholding structure of SOEs to regress the full sample. Table 4 reports the test results. Dependent variable is corporate governance level (*gover*). Independent variable is ownership structure (*gover*). From column (1), it can be seen that there is a positive relationship between the governance of non-state shareholders and the level of governance. Additional, whether it is a simple OLS regression (coefficient of *nsta* in column (2) is 0.174), random effects of the panel (coefficient of *nsta* in column (3) is 0.133) or fixed effects (The coefficient of nsta in column (4) is 0.094), or in the analysis of the cluster robust standard error regression after

**Table 3. Correlation coefficient table.**

| Variable | *gover* | *nsta* | *pai_nsta* |
|----------|---------|--------|------------|
| *gover* | 1 | 0.0361*** | 0.0621*** |
| *nsta* | 0.0542*** | 1 | 0.4875*** |
| *pai_nsta* | 0.0667*** | 0.6126*** | 1 |

**Table 4. Shareholding structure and corporate governance level.**

| | corporate governance level: *gover* | | | | | | | |
|---|---|---|---|---|---|---|---|---|
| | 1.OLS | 2.RE | 3.FE | 4.Cluster | 1.OLS | 2.RE | 3.FE | 4.Cluster |
| nsta | 0.174*** | 0.133*** | 0.094** | 0.133** | | | | |
| | (4.69) | (3.24) | (1.98) | (2.26) | | | | |
| pai_nsta | | | | | 0.361*** | 0.472*** | 0.519*** | 0.361*** |
| | | | | | (22.96) | (27.12) | (27.03) | (17.85) |
| size | -0.132*** | -0.110*** | -0.094*** | -0.110*** | -0.113*** | -0.084*** | -0.084*** | -0.113*** |
| | (-16.16) | (-8.61) | (-4.77) | (-6.17) | (-14.65) | (-6.97) | (-4.71) | (-13.32) |
| manage | 3.485*** | 4.398*** | 6.870*** | 4.398*** | 2.124*** | 2.925*** | 4.983*** | 2.124*** |
| | (11.25) | (13.61) | (12.36) | (3.83) | (7.24) | (9.59) | (9.88) | (4.14) |
| lev | 0.032 | -0.031 | -0.082 | -0.031 | 0.066 | -0.035 | -0.097 | 0.066 |
| | (0.54) | (-0.46) | (-1.03) | (-0.36) | (1.18) | (-0.55) | (-1.35) | (1.16) |
| roa | -1.642*** | -0.734*** | -0.560*** | -0.734*** | -1.574*** | -0.745*** | -0.607*** | -1.574*** |
| | (-7.62) | (-4.05) | (-2.97) | (-3.42) | (-7.78) | (-4.50) | (-3.55) | (-7.36) |
| cfo | -0.007 | -0.064 | -0.036 | -0.064 | -0.035 | -0.061 | -0.040 | -0.035 |
| | (-0.05) | (-0.58) | (-0.32) | (-0.50) | (-0.26) | (-0.60) | (-0.39) | (-0.26) |
| grow | 0.001 | 0.002 | 0.002 | 0.001 | 0.002 | 0.003 | 0.004 | 0.002 |
| | (0.17) | (0.56) | (0.62) | (0.46) | (0.56) | (1.16) | (1.52) | (0.86) |
| expend | -0.498** | -0.530*** | -0.533*** | -0.530*** | -0.465** | -0.507*** | -0.482*** | -0.465** |
| | (-2.38) | (-2.91) | (-2.81) | (-2.58) | (-2.36) | (-3.05) | (-2.81) | (-2.45) |
| _cons | 2.526*** | 1.901*** | 1.569*** | 1.901*** | -0.010 | 0.065 | -0.140 | 0.065 |
| | (7.73) | (5.71) | (3.36) | (3.91) | (-0.08) | (0.46) | (-0.65) | (0.60) |
| year & firm | YES | YES | YES | YES | YES | YES | YES | YES |
| N | 4101 | 4101 | 4101 | 4101 | 4101 | 4101 | 4101 | 4101 |
| R² | 0.1422 | 0.2513 | 0.2163 | 0.2513 | 0.2421 | 0.2817 | 0.2584 | 0.2421 |

***, **, and * denote significance at the 1%, 5%, and 10% level, respectively. Due to amplitude limitations, the subsequent empirical results only display the analysis of core variables. Taking into account the heterogeneity among different individuals, the subsequent empirical analysis model will report t-values based on robust standard errors clustered at the enterprise level.

considering heteroscedasticity (coefficient of *nsta* in column (5) is 0.133), the non-state shareholder variable (*nsta*) at the level of shareholding structure both are significantly positive, which means that the introduction of non-state shareholders in SOE has a positive impact on the governance level. Hypothesis H1A is verified. When the board structure is used as an independent variable, the coefficient *pai_nsta* is significantly positive. This means that non-state-owned directors can help improve the governance level of state-owned enterprises. Hypothesis H1B is verified.

## The instrumental variables method

While we document a positive association between non-state shareholder governance and governance level of SOE, our results may be subject to self-selection bias. That is, SOE with a high level of governance means that the corporate governance mechanism is relatively complete, and it is more likely to be favored by non-state shareholders. In a word, the decision to involve non-state shareholders in the governance of SOEs is not a random choice, which can introduce self-selection bias. In addition, given the availability of data in the sample, the selection of variables tends to bias the sample. We may still suffer from omitting unknown variables that affect non-state shareholder governance and governance level of SOE, and produce a more severe endogeneity bias. To mitigate this potential issue, we use the Instrumental Variables Method.

Our first instrument stems from policy characteristic. According to the policy document "Guidance on the Functional Definition and Classification of State-owned Enterprises" issued

by SASAC, different types of SOEs should be classified to promote reform and actively introduce other capital to diversify the equity of SOEs following the requirements of different industries. Under this exogenous policy, with reference to Xiong and Zhang (2020) [8], we then select the mean of heterogeneous shareholder ownership as instrumental variables (*meannsta* and *meanpai_nsta*).

Our second instrument stems from historical characteristic. Conceptually, for two SOEs with the same size, the SOE in waterfront countries or regions are more likely to interact with companies in other countries, so the region where the SOE is located is closely linked to its institutional development. Therefore, we would like to find an instrument that affected mixed ownership reform of SOE but did not affect governance level of SOE through any other channel. Considering the policy of mixed ownership reform, the extent of non-state shareholders' entry into SOEs is influenced by the government's willingness to decentralize power, we refer to the practice of Fan et al (2013) [58], using the first Opium War to the founding of New China whether forced to open as a commercial port or set up a concession (*com_port*) as an instrumental variable. From the First Opium War to before the founding of New China (1842–1949), the Chinese government was forced to open trade ports (ie commercial ports) and set up concessions, thus strengthening exchanges with other countries. According to Fan et al. (2013) [58], these trading ports include: Fujian, Guangdong, Shanghai, Zhejiang, Hainan, Hubei, Jiangsu, Liaoning, Shandong, Tianjin, Xinjiang, Anhui, Guangxi, Chongqing and Hebei; concessions include: Tianjin, Jiangsu, Shanghai, Zhejiang, Anhui, Jiangxi, Fujian, Guangdong, Shandong, Chongqing and Hubei. The overlap of the two is high, so the union is taken as the instrument variable indicator. Before the First Opium War, due to the isolation and isolation of the Qing Dynasty, China interrupted exchanges with other countries, and a series of agreements and treaties signed after that required the Chinese government to open some trade ports and set up concessions to allow foreign capital to invest and set up factories economic and cultural exchange activities such as mission, mission and school opening. Therefore, the system construction and reform of SOEs in commercial ports and concession areas may be relatively complete, and governance level of SOE has little to do with whether the area is a commercial port or a concession. Therefore, whether an area is a commercial port and whether it is a concession satisfy the instrumental variables of the two standards. We adopt the 2SLS approach to reduce the effect of endogeneity issues on the test results.

The results of this analysis are reported of Table 5 (first column) presents the first stage results. The coefficient estimates on *meannsta/ meanpai_nsta* and *com_port* across all first-stage regressions are positive and significant at the 1% level, suggesting that our IVs are positively associated with non-state shareholder governance. Since the P-value of F-test, i.e. 0.000 of the first-stage regressions is significant at the 1% level, we can reject the null hypothesis that these instruments are weak. Therefore, the coefficient estimates and their corresponding t-statistics in the second stage are likely to be unbiased and inferences based on them are reasonably valid. Then, the insignificant value of Sargan test (p = 0.2154) infers that our instrument is not over-identified. Taken together, these tests indicate the selected instrument is correctly identified, strong and valid.

We then present the second-stage regression results in columns (2) of Table 5. The coefficient estimates on the non-state shareholder governance (*nsta/ pai_nsta*) are positive and statistically at the 1% level across both columns. The IV regression results are consistent with our baseline findings and further support our predictions that non-state shareholder governance improved governance level of SOE.

**Table 5. Instrumental variable approach.**

| | corporate governance level: *gover* | | | |
| --- | --- | --- | --- | --- |
| | 1<sup>st</sup>-stage | 2<sup>nd</sup>-stage | 1<sup>st</sup>-stage | 2<sup>nd</sup>-stage |
| | *nsta* | *gover* | *pai_nsta* | *gover* |
| *nsta* | | 0.081*** | | |
| | | (4.37) | | |
| *meannsta* | 0.318*** | | | 0.069*** |
| | (6.57) | | | (3.98) |
| | | | 0.147*** | |
| | | | (4.20) | |
| *com_port* | 0.291*** | | 0.121*** | |
| | (3.96) | | (5.37) | |
| *_cons* | 0.157*** | 3.710*** | 0.284*** | 5.980*** |
| | (3.46) | (4.35) | (5.68) | (3.72) |
| *controls* | YES | YES | YES | YES |
| *year & firm* | YES | YES | YES | YES |
| *N* | 4101 | 4101 | 4101 | 4101 |
| *R²* | 0.025 | 0.473 | | |
| First stage F test | 157.25*** | | 158.17*** | |
| Sargan statistic(p-value) | 0.2154 | | 0.2685 | |

## Robustness tests

**Propensity score matching (PSM).**   Although we control for these characteristics in our linear OLS model, the model is not adequate for controlling potential non-linear effects. The high level of SOE governance may be attributed to differences in characteristics between whether there are board members with non-state background of SOEs. As a robustness check, we re-run our model a propensity score matched sample to mitigate this concern. Specifically, we use logit model to regress our indicator variable board governance (*pai_nsta*) on control variables in model (1) and estimate the propensity score that a SOE have a board member with non-state background. Secondly, we match each treatment SOE (Heterogeneous background = 1) with a control SOE (Heterogeneous background = 0) with the closest propensity score. We use a one-to-one no-replay PSM approach. We re-estimate the regression model using matched sample, and the results shown in Table 6 column (1) indicate that our main findings are robust. The coefficients of Heterogeneous background remains positive and statistically significant at the 1% level. In summary, the finding is unlikely to be driven by inadequate control of non-linearity.

**Alternative measure of independent variable.**   In column (2) of Table 6, we present results from using change analysis by converting our main independent variables, namely the shareholding structure (*nsta*) and board structure (*pai_nsta*) into heterogeneous equity integration (*equity*) and heterogeneous director (*n_dummy*). First, we calculate the proportion of state-owned shares and non-state shares (the sum of the shareholdings of shareholders of other nature than state-owned shareholders) to the total equity in SOEs as Es and Ep, respectively, and use the larger of *Es* and *Ep* as the denominator and the smaller as the numerator (i.e., *equity* = Ep/Es when *Es>Ep* and *equity* = Es/Ep when *Es<Ep*), and define the resulting ratio as the heterogeneous equity integration. If there are director members appointed by non-state shareholders in the SOE, then *n_dummy* is 1, and otherwise it is 0. The results reveal consistent findings across all the models. These results imply that the relationship between non-

**Table 6. Robustness tests.**

| | corporate governance level: *gover* | | | | | | |
|---|---|---|---|---|---|---|---|
| | **(1) board structure** | **(2)** | | **(3)** | | **(4)** | |
| | PSM | Alternative Measure | | 0–1 variable | | Change time samples | |
| *nsta* | | | | 0.098*** (3.17) | | 0.151** (2.43) | |
| *pai_nsta* | | | | | 0.723*** (18.43) | | 0.474*** (14.17) |
| *treat* | 0.394*** (3.81) | | | | | | |
| *equity* | | 0.114*** (3.78) | | | | | |
| *n_dummy* | | | 0.567*** (25.64) | | | | |
| *_cons* | 13.09*** (6.98) | 1.888*** (2.60) | 0.502 (1.16) | 2.564*** (10.57) | 0.924*** (2.59) | 2.193*** (5.72) | 1.246*** (2.94) |
| controls | YES | YES | YES | YES | YES | YES | YES |
| year & firm | YES | YES | YES | YES | YES | YES | YES |
| N | 2386 | 4101 | 4101 | 4097 | 4097 | 2324 | 2324 |
| $R^2$ | 0.3844 | 0.2411 | 0.3907 | 0.0789 | 0.2423 | 0.2423 | 0.2642 |

state shareholder governance and corporate governance level is not sensitive to model specification.

**Dependent variable.** We use a 0–1 dummy variable to measure the SOE governance level. Specifically, we take the median of the governance level variable as the boundary, and if it is less than the median, it is 0, otherwise it is 1. We re-analyze the model empirically. The results are shown in column (3) of Table 6, the significance of the coefficient is consistent with the above empirical results.

**Change time sample.** Mixed ownership reform of SOE was proposed in the 1990s but was not substantially promoted until the Third Plenary Session of the 18th Central Committee in 2013 when non-state shareholder governance took the stage in the management of SOEs. In column (4) of Table 6, we exclude data from the sample prior to 2013 and re-estimated the model. The relationship between non-state shareholder governance and governance level of SOE remains consistent after dropping sample data before 2013.

## Potential channel: Board resolution behavior

In this sub subsection, we explore potential mechanism through which non-state shareholder governance positively affects governance level of SOE: Board Resolution Behavior.

We expect directors appointed by non-state shareholders to increase the level of board dissent. If that is the case, we test this mechanism by examining whether non-state shareholders governance are more likely to increase the level of board dissent. Based on the upper echelons theory, the heterogeneous background of board personnel can lead to disagreement [47]. Gomes and Novaes (2005) argue that directors appointed by non-state shareholders are able to participate in board resolutions by "voting with their hands" and bring differentiated and rational suggestions to board motions through a market-oriented and professional perspective [50], effectively monitoring and modifying the management activities and decision-making behavior of SOEs, thus enhancing the effectiveness of SOE decision-making. Therefore, "principled dissent" promotes business growth [51].

To test this prediction, we follow prior literature and use *dissent* as independent variable. Table 7 reports the results. Consistent with our expectation, we find that non-state shareholder governance is positively associated with the likelihood of obtaining board dissent. The coefficients (*nsta* or *pai_nsta*) at significant at 10% level, which indicates non-state shareholder governance at both the shareholding structure and senior governance levels has increased the positivity of board resolutions. Hypothesis 2 is verified. To keep the results robust, we further use logit model to run the regression model. The empirical results are shown in Table 7 column (2), and the results are consistent with the previous findings.

## The regulating function of party organization governance

The party organization is a SOE governance system with Chinese characteristics. Whether there are important party organization members on the board of directors may lead to heterogeneous results of mixed ownership reform. This section will empirically test H3, that is, the influence of the party organization on the relationship between the governance of non-state shareholders and the level of governance. To test this hypothesis, we divided the entire sample into two groups: party = 0 group and party = 1 group, and further conducted group regression analysis.

We conduct group regression to examine the impact of non-state shareholder governance on SOE with or without important party members on the board of directors. The regression results are shown in Table 8. Dependent variable is corporate governance level (*gover*). The study found that when there is no important member of the party organization in the board of directors (Columns 1 to 4), whether it is *nsta* or *pai_nsta*, the governance of non-state shareholders is positively significant at the level of 10%. This means that the governance of non-state shareholders actively promotes the governance of SOE. This may be because non-state shareholders are more active in participating in SOE governance when there are no important members of the party organization on the board of directors. And when there are important members of the party organization in the board of directors (Columns 5 to 6), the governance of non-state shareholders (*nsta* and *pai_nsta*) is positive but not significant. Hypothesis 3 is verified. Of course, this may be related to the fact that SOEs are affected by the governance of the party organization, and it is difficult to undergo significant changes due to the governance of a single non-state shareholder. To ensure comparability between the two groups, we further adopt the propensity score matching (nearest neighbor matching) method, taking *party* = 1 as the experimental group, and vice versa. The empirical results show that it remains robust.

**Table 7. Potential channels.**

|  | *dissent* | | | |
| --- | --- | --- | --- | --- |
|  | **(1)** | | **(2) logit model** | |
| *nsta* | 0.047** (2.22) |  | 1.000** (2.56) |  |
| *pai_nsta* |  | 0.0148* (1.74) |  | 0.0148* (1.74) |
| *_cons* | 0.0993 (1.19) | 0.0828 (0.97) | -2.708 (-1.22) | -3.253 (-1.45) |
| *controls* | YES | YES | YES | YES |
| *year & firm* | YES | YES | YES | YES |
| N | 4101 | 4101 | 4097 | 4097 |
| $R^2$ | 0.0308 | 0.0271 | 0.0173 | 0.0248 |

**Table 8. Group test of corporate governance level.**

| | corporate governance level: *gover* | | | | | |
| | party = 0 | | | | party = 1 | |
| | **(1)** | **(2)** | **(3) PSM** | **(4) PSM** | **(5)** | **(6)** |
|---|---|---|---|---|---|---|
| *nsta* | 0.124*<br>(1.68) | | 0.013***<br>(3.94) | | 0.119<br>(0.64) | |
| *pai_nsta* | | 0.485***<br>(11.99) | | 0.538**<br>(2.24) | | 0.509***<br>(4.40) |
| *_cons* | 1.544**<br>(2.09) | 1.071<br>(1.46) | 3.369***<br>(13.28) | 2.871***<br>(14.68) | 0.344<br>(0.15) | -0.029<br>(-0.01) |
| *controls* | YES | YES | YES | YES | YES | YES |
| *year & firm* | YES | YES | YES | YES | YES | YES |
| N | 3797 | 3797 | 3140 | 3140 | 304 | 304 |
| $R^2$ | 0.2280 | 0.2765 | 0.2933 | 0.2049 | 0.1047 | 0.2786 |

Furthermore, we used board resolution behavior (*dissent*) as the dependent variable to examine the behavioral effect of non-state shareholders when there are important party members in the board. Table 9 reports the results of the analysis. The results show that when *party* = 0, the governance of non-state shareholders (*nsta* and *pai_nsta*) has a significant positive impact on the board resolution behavior; while when *party* = 1, the governance of non-state shareholders (*nsta* and *pai_nsta*) is not significant. However, there may still be a problem of self-selection caused by the fact that the grouping variables are not exogenous. We further employed propensity score matching method, and the empirical results still remain robust (columns 3 to 4).

We found that when the party organization does not participate in board governance, non-state shareholders, as "outsiders", can actively participate in board governance, express their opinions on board proposals that are not conducive to enterprise development, and actively perform decision-making and supervisory functions. When the party organization participates in the governance of the board of directors, although it can ensure the decision-making and supervision functions of the board of directors [54], it cannot mobilize the participation of non-state shareholders or directors in the resolution process of the board of directors. To sum up the research conclusions, although some scholars have verified the positive governance effect of the party organization's "discussion before" decision-making mechanism, from the point of view of the decision-making behavior of the board of directors, the party organization

**Table 9. Group test of board resolution behavior.**

| | board resolution behavior: *dissent* | | | | | |
| | party = 0 | | | | party = 1 | |
| | **(1)** | **(2)** | **(3) PSM** | **(4) PSM** | **(5)** | **(6)** |
|---|---|---|---|---|---|---|
| *nsta* | 0.048**<br>(2.17) | | 0.079***<br>(3.34) | | 0.037<br>(0.62) | |
| *pai_nsta* | | 0.016*<br>(1.86) | | 0.030**<br>(2.24) | | 0.019<br>(0.60) |
| *_cons* | 0.125<br>(1.51) | 0.108<br>(1.30) | 0.093<br>(0.38) | 0.088<br>(0.36) | -0.330<br>(-0.90) | -0.017<br>(-0.01) |
| *controls* | YES | YES | YES | YES | YES | YES |
| *year & firm* | YES | YES | YES | YES | YES | YES |
| N | 3797 | 3797 | 3140 | 3140 | 304 | 304 |
| $R^2$ | 0.0348 | 0.0328 | 0.0452 | 0.0563 | 0.0668 | 0.0701 |

governance inhibits the e participation of non-state directors in the process of board governance.

## The governance effect of non-state shareholders based on the resolution behavior of the board of directors

The above empirical analysis shows that the governance of non-state-owned shareholders can increase the participation of directors in the board of directors' decision-making process, but whether this behavior will affect the governance level of SOEs is still unclear. On this basis, the following empirical model is constructed:

$$gover_{it} = \alpha_0 + \alpha_1 nsta_{it}/pai\_nsta_{it} + \alpha_2 dissent_{it} + \alpha_3 nsta_{it}/pai\_nsta_{it}$$
$$\times dissent_{it} + \sum controls + year + \mu_i + \varepsilon_{it} \tag{2}$$

Among them, *gover* represents the governance level of state-owned enterprises. In order to eliminate the collinearity after adding the interaction term, the *nsta/pai_nsta* and *dissent* are separately processed.

Table 10 reports the results. In column (1), the party organization variable (*party*) is significantly positive. This implies that party organizational governance can enhance the governance level of state-owned enterprises. In columns (2) and (3), non-state shareholder governance variables (*nsta* and *pai_nsta*) and board resolution variables are both significantly positive, while the interaction terms *(nsta×dissent* and *pai_nsta×dissent)* are not significant. Furthermore, we divide the entire sample into boards with no significant party members (= 0) and boards with significant party members (*party* = 1). Columns (4) and (5) show that when *party* = 0, the coefficient of the interaction term (*nsta×dissent* and *pai_nsta×dissent*) is significantly negative. This indicates that in state-owned enterprises without party organizational governance, the board decision-making behavior of non-state shareholders/directors actually reduces the corporate governance level. When *party* = 1, columns (6) and (7) reveal that the

**Table 10. Test of governance effect of non-state shareholders based on board resolutions.**

| | corporate governance level: *gover* | | | | | | |
|---|---|---|---|---|---|---|---|
| | *full sample* | | | *party = 0* | | *party = 1* | |
| | **(1)** | **(2)** | **(3)** | **(4)** | **(5)** | **(6)** | **(7)** |
| *party* | 0.097*<br>(1.73) | | | | | | |
| *nsta* | | 0.139**<br>(1.97) | | 0.137*<br>(1.83) | | 0.086<br>(0.47) | |
| *pai_nsta* | | | 0.513***<br>(12.75) | | 0.490***<br>(12.26) | | 0.746***<br>(8.67) |
| *dissent* | | 0.119**<br>(1.99) | 0.120**<br>(2.18) | 0.162**<br>(2.46) | 0.169***<br>(2.79) | -0.102<br>(-0.94) | -0.056<br>(-0.31) |
| *nsta×dissent* | | -0.204<br>(-1.32) | | -0.300*<br>(-1.90) | | 1.274**<br>(2.35) | |
| *pai_nsta×dissent* | | | -0.112<br>(-1.47) | | -0.165**<br>(-2.16) | | 0.126<br>(0.52) |
| *_cons* | 1.536**<br>(2.14) | 1.587**<br>(2.23) | 0.939<br>(1.33) | 1.561**<br>(2.12) | 1.062<br>(1.45) | 1.245<br>(0.54) | 0.221<br>(0.11) |
| *controls* | YES | YES | YES | YES | YES | YES | YES |
| *year & firm* | YES | YES | YES | YES | YES | YES | YES |
| N | 4101 | 4101 | 4101 | 3797 | 3797 | 304 | 304 |
| $R^2$ | 0.2158 | | | 0.2289 | 0.2203 | 0.1166 | 0.3753 |

interaction term (*nsta×dissent* and *pai_nsta×dissent*) coefficients are significantly positive. This suggests that in the board resolution process, non-state shareholder governance contributes to enhancing the governance level. Hypothesis 4 is confirmed. The results indicate that in the field of corporate governance, the introduction of non-state shareholders/directors in SOEs needs to be complemented with party organizational governance to fully leverage the governance advantages of non-state shareholders.

Based on the upper echelons theory, the board of directors plays the role of automatic error correction in corporate governance, which is conducive to improving the governance level of SOE. The party organization governance can more effectively exert the effect of the mixed ownership reform policy, guide the positive behavior of directors in the resolutions of the board of directors, supervise and make decisions on the operation and development of SOE, and help optimize the internal governance mechanism and promote the development of SOE. Sustainable and healthy development.

## Conclusions and implications

### Conclusions

Under the background of deepening the reform of SOE, in order to adhere to the "two consistency" policy guidelines of SOE and realize the structural adjustment of the governance mechanism of SOE, we explore the impact mechanism of non-state shareholders on the governance level of SOE from the perspective of board resolutions and party organization governance. We found that governance by non-state shareholders improves the governance of SOEs. Moreover, the resolution of the board of directors exerts an intermediary effect, that is, the governance of non-state shareholders has the courage to raise objections to the board of directors, giving full play to the effectiveness of the resolution of the board of directors. Furthermore, we found that when party members serve as directors, non-state shareholder governance has a significantly positive impact on the governance level of SOEs.

### Theoretical implications

Based on Chinese mixed ownership reform system, we discuss the impact of non-state shareholder governance on the governance level of state-owned enterprises, breaking the question raised by Goranova and Ryan (2022) whether heterogeneous shareholders can improve the effectiveness of corporate governance, and enriching the research in the field of corporate governance. Related research [4]. The specific contributions are as follows.

First, by focusing on SOE in China, we expand on recent discussions on the effects of mixed ownership reforms. Previous research on the impact of non-state shareholder governance practices has confirmed that non-state shareholder governance has a positive impact on SOE in general. Most of these quantitative studies are based on the mix of ownership structures, follow the classical resource dependence theory [54], and emphasize the importance of resource complementarity. Different from the resource effect of non-state shareholders, we explore how the governance of non-state shareholders affects the governance behavior of SOE. Thus, we reveal the impact of non-state shareholder governance on the governance level of SOE, and provide a more comprehensive understanding of the effects of mixed ownership reform in SOE.

Second, by investigating the mechanism of the impact of non-state shareholder governance on SOE governance, we expand the research on the motivation of non-state shareholder governance on SOE governance. However, the existing literature mostly discusses the governance effect of non-state shareholders in SOE with the help of resource dependence theory and principal-agent theory. From the perspective of high-level ladder theory, we explore the

governance behavior of non-state shareholders participating in SOE board resolutions. Considering the historical problems of "lack of owner" and "insider control" in SOE, shareholder centralism is not suitable for corporate governance of SOE. Therefore, we confirm that, with board resolutions as an intermediate link, SOE can adopt board-centrism to realize modern corporate governance with Chinese characteristics, making the research on corporate governance of SOE more concrete.

Third, we also contribution to the SOE literature by introducing party organization governance as boundary condition. Previous research has shown that the corporate governance mechanism of SOE is affected by the governance of the party organization [43, 54]. According to the stakeholder theory, non-state shareholders and party organization members are all stakeholders of SOE. However, few scholars have paid attention to whether the governance of the party organization will affect the governance effect of the mixed ownership reform of SOE. By taking China as the research background, we capture the moderating role of party organization governance, thereby observing the different responses of non-state shareholder governance to the governance level of SOE. Thus, we have worked on connecting mixed ownership reform and party organization governance, as well as board resolution, to enrich the research context on SOE corporate governance.

## Practical implications

Combined with theoretical analysis and empirical research conclusions, the research in this paper has certain practical significance. For a long time, the outside world has had mixed opinions on the mixed-ownership reform of SOE; at the same time, due to the blurred boundaries of party organization governance, enterprises cannot fully exert the enthusiasm of various governance subjects in corporate governance. This study reflects the relationship between non-state shareholders and party organizations from the level of board governance, and the conclusion is crucial to improving the governance capacity of SOE. The research results of this paper provide more specific policy implications for deepening the reform of SOE and clarifying the internal governance relationship of enterprises.

First of all, we emphasize the positive impact of non-state shareholder governance on the governance level of state-owned enterprises, offering the potential for increased economic vitality and market competitiveness. By reinforcing the role of board resolutions, the governance capacity of non-state shareholders has been proven to enhance the overall governance quality of state-owned enterprises, providing a more effective decision-making mechanism for business development. Hence, it is necessary to focus on "reforming" the corporate governance mechanism, giving non-state shareholders or directors real decision-making power and voice, establishing a good governance mechanism, promoting the transformation of SOE from shareholder governance to board governance, and ensuring that the board of directors normally exercise supervision With the function of decision-making, improve enterprise vitality and market competitiveness.

Secondly, it is necessary to encourage directors to actively express their opinions in the resolutions of the board of directors, and make suggestions for the management and governance of the enterprise in the spirit of "ownership", so as to promote the long-term development of the enterprise.

Finally, considering that party organizational governance contributes to unleashing the governance effects of non-state shareholders, state-owned enterprises should strengthen party organizational governance and delineate decision-making responsibilities between market and non-market strategies. Alternatively, they may reconfigure the "prepositioning" of party organizational governance as a "post-positioning" approach, placing its decision-making role as the

ultimate guardian at the end. This approach does not, however, dampen the enthusiasm of non-state shareholders in the board resolution process.

## Limitations and future research directions

This study has several limitations. Firstly, it primarily focuses on the mixed ownership reform of Chinese state-owned enterprises, thus limiting the generalizability of the conclusions to other regions. Secondly, the research relies predominantly on quantitative methods and lacks a diverse methodological approach, such as qualitative studies and case analyses, potentially hindering a comprehensive understanding of the impact mechanisms of non-state shareholder governance. Additionally, the study's timeframe extends only until 2019, failing to capture the dynamic processes of the mixed ownership reform comprehensively and potentially overlooking recent impacts of governance reforms.

Future research can explore several directions: Firstly, there is a need to deepen the study of governance mechanisms, with a focus on understanding the specific impact of non-state shareholders on the operational aspects of governance mechanisms within state-owned enterprises. Secondly, investigating the variations in the governance effectiveness of non-state shareholders across different industries and company sizes would provide valuable insights. Thirdly, conducting more longitudinal studies to track the long-term effects of governance reforms would enhance our understanding of the sustained impact of these reforms. Lastly, extending the scope of research to include other countries and comparing governance models in different nations would offer comparative insights into state-owned enterprise reforms globally. By addressing these research avenues, future studies can achieve a more comprehensive understanding of the impact of mixed ownership reforms on the governance of state-owned enterprises, providing more targeted recommendations and policy support.

## Acknowledgments

The authors would like to thank the editor and the reviewers for the insightful comments to improve the manuscript.

## Author Contributions

**Conceptualization:** Aihua Xiong.

**Data curation:** Zhibin Zhang.

**Funding acquisition:** Aihua Xiong.

**Methodology:** Zhibin Zhang.

**Validation:** Lishu Zhang.

**Writing – original draft:** Zhibin Zhang.

**Writing – review & editing:** Lishu Zhang.

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
