## [Decision Letter · Decision Letter 0]

2 Jan 2024

PONE-D-23-33605The Influence of Non-state Shareholder Governance on the Governance Level of State-owned Enterprises——Based on the Perspective of Board Resolution Behavior and Party Organization GovernancePLOS ONE

Dear Dr. Zhang,

Thank you for submitting your manuscript to PLOS ONE. After careful consideration, we feel that it has merit but does not fully meet PLOS ONE’s publication criteria as it currently stands. Therefore, we invite you to submit a revised version of the manuscript that addresses the points raised during the review process.

The paper investigate The Influence of Non-state Shareholder Governance on the Governance Level of State-owned Enterprises——Based on the Perspective of Board Resolution Behavior and Party Organization Governance". Its findings are interesting but requires minor revisions before it can be considered. My comments are as follows:

You need state clearly the contributions of the paper. For example, "Consequently, the current paper seeks to make the following contributions to the existing literature. First,…, Second,…., Third, …, Fourth,… and so on". The description of the contribution needs to be more forensic, needs to be more focussed.Highlight their economic and research and policy implications. In the discussion of the results please focus on the novel findings and insights vis-à-vis the existing literaturePlease submit your revised manuscript by Feb 16 2024 11:59PM. If you will need more time than this to complete your revisions, please reply to this message or contact the journal office at plosone@plos.org. Please include the following items when submitting your revised manuscript:A rebuttal letter that responds to each point raised by the academic editor and reviewer(s). You should upload this letter as a separate file labeled 'Response to Reviewers'.A marked-up copy of your manuscript that highlights changes made to the original version. You should upload this as a separate file labeled 'Revised Manuscript with Track Changes'.An unmarked version of your revised paper without tracked changes. You should upload this as a separate file labeled 'Manuscript'.If applicable, we recommend that you deposit your laboratory protocols in protocols.io to enhance the reproducibility of your results. Protocols.io assigns your protocol its own identifier (DOI) so that it can be cited independently in the future. For instructions see: https://journals.plos.org/plosone/s/submission-guidelines#loc-laboratory-protocols. Additionally, PLOS ONE offers an option for publishing peer-reviewed Lab Protocol articles, which describe protocols hosted on protocols.io. Read more information on sharing protocols at https://plos.org/protocols?utm_medium=editorial-email&utm_source=authorletters&utm_campaign=protocols.

We look forward to receiving your revised manuscript.

Kind regards,

Wajid Khan

Academic Editor

PLOS ONE

“This work was supported by the Key Projects of Shandong Provincial Social Science Planning, Research Grant number 21BGLJO2，Shandong Technology and Business University Talent Introduction Project.”

Reviewers' comments:

Reviewer's Responses to Questions

**Comments to the Author**

1. Is the manuscript technically sound, and do the data support the conclusions?

Reviewer #1: Yes

Reviewer #2: Yes

2. Has the statistical analysis been performed appropriately and rigorously? 

Reviewer #1: Yes

Reviewer #2: Yes

3. Have the authors made all data underlying the findings in their manuscript fully available?

Reviewer #1: Yes

Reviewer #2: Yes

4. Is the manuscript presented in an intelligible fashion and written in standard English?

Reviewer #1: Yes

Reviewer #2: Yes

5. Review Comments to the Author

Reviewer #1: This paper takes manufacturing SOEs in the competitive field from 2009 to 2019 as the research sample, and explores the impact of mixed-ownership reform on the governance level of SOEs from the perspectives of shareholding structure and board structure. The paper is based on proper data and methods, but there are several issues which should be revised.

1. The abstract should clearly explain the research motivation, content, methods, conclusions, and significance of the article. Please modify and improve the abstract.

2. From the hypothesis, it can be seen that the author divides the governance of non-state-owned shareholders into shareholding structure and board structure, and proposes hypotheses separately from two aspects. However, we do not see the logical differences in the impact of non-state-owned shareholders' shareholding and appointing directors on the corporate governance level in this section, which are explained from reducing agency costs, exerting supervision and balance. It is recommended to explore the impact paths of shareholding structure and board structure on the corporate governance level in a targeted manner. The current analysis is not sufficient.

3. Why is the sample limited to state-owned manufacturing enterprises in the competitive field? Many studies on non-state-owned shareholder governance are based on a full sample, and the article does not explain the reasons for selecting the sample. The research background discussed in the introduction and the subsequent hypothesis proposed did not focus on specific samples.

4. It is recommended to place the empirical model after the "Variable Selection" section, rather than in the empirical results section.

5. There are certain issues with the empirical testing of this article. The " Theoretical analysis and research hypothesis" section elaborates on the impact of shareholding structure and board structure, and suggests testing both aspects simultaneously in the "Baseline regression results" section, rather than focusing solely on shareholding structure.

6. The limitations of the research and future research directions should be explained in the conclusion of the article.

Reviewer #2: 1) Problem statement of the study must be supported by most recent literature

2) Significance of study must be specified in introduction section.

3) Most recent citations must be provided in literature section. It was difficult to find any citation from 2022 and 2023, authors are suggested to add literature from 2022 and 2023 to support research objectives and research hypothesis.

4) A short review of governance policies implemented in China must be explained to provide readers an insight about it.

5) All variables of the study should be further explained in a table to make it more convenient for readers to understand study variables.

6) Sampling technique for sample selection is missing in research methodology section. Provide sampling technique with justification

7) Data for study is collected from 2009 to 2019. It is suggested to increase the sample time period upto 2022 to make the study more authentic.

8) Statistical techniques used in the study must be explained in research methodology section along with all required quantitative models or equations.

9) All results in data analysis section must be supported by similar studies carried out in China or in the Asian region

6. PLOS authors have the option to publish the peer review history of their article (what does this mean?). If published, this will include your full peer review and any attached files.

Reviewer #1: No

Reviewer #2: No

---

## [Author Response · Author response to Decision Letter 0]

23 Feb 2024

We sincerely appreciate the editor's very kind and careful examination of our paper. The reviewers’ comments are very right and helpful to us. We are very grateful to the reviewers for their hard work and the important comments for improving the quality of our article.

We have studied all the comments and made revisions very carefully. The following is an outline of the revisions that we have made.

Response to Editor

1. You need state clearly the contributions of the paper. For example, "Consequently, the current paper seeks to make the following contributions to the existing literature. First,…, Second,…., Third, …, Fourth,… and so on". The description of the contribution needs to be more forensic, needs to be more focussed.

1. We are very grateful to the editor for their suggestions and related reviews. Based on the editor's suggestions, we have revised and improved the contributions of the paper as follows (the modified content is at the end of the introduction):

Consequently, the current paper seeks to make the following contributions to the existing literature. Firstly, the study deepens the understanding of mixed ownership reform, revealing its mechanisms and effects in enhancing the governance efficiency of SOEs. First, we add to the literature documenting the economic benefits of non-state shareholder governance in SOEs. Several recent studies have examined the impacts of non-state shareholder governance on innovative decision-making (Wang et al., 2021), operating performance (Guan et al., 2021), and technology innovation (Zhang et al., 2020). Research on non-state shareholder governance and governance level of SOE, what is not been thoroughly examined in detail thus far are the reasons for such outcomes. In addition, although there are many literatures on the level of corporate governance, most of them study corporate governance as a pre-variable (Ammann et al., 2011; Fu and Hong, 2018; Chu et al., 2019). Taking Zhejiang enterprises as the research object, Du et al. (2021) analyzed the governance effect evaluation of the mixed ownership reform, but there is insufficient research on the improvement path of its governance capacity. We take corporate governance level as a post-variable It is very important in SOE to explore the impact of non-state shareholder governance on governance level. We complement these studies by providing evidence that non-state shareholder governance promotes governance level of SOE.

Further, we identify the mechanisms of non-state shareholders on governance level from the perspective of board resolutions. One of the most critical aspects of our results is that, in addition to highlighting the influence of non-state shareholder governance on governance level, we also represent indirect evidence of the influence of non-state shareholder governance and board resolutions on governance level of SOE. Thus, our results highlight how non-state shareholder governance interacts with board resolutions and the governance level.

Secondly, by means of policy measures, the study provides empirical support for heterogeneous shareholder governance, bridging the gap between theory and practice. Second, we place party organization governance within the framework of non-state shareholder governance and SOE governance. Party organization and mixed ownership reform are unique systems of Chinese SOE, and most of the relevant literature studies them separately. As important stakeholders, whether important members of the party organization and non-state shareholders/directors can exert the governance effect of "1+1>2" remains to be tested. Therefore, combined with the mixed ownership reform system, we incorporate the party organization into the analytical framework of corporate governance at the micro level of SOE, explore the relationship between non-state shareholders and party organization governance, and provide theoretical and empirical support for it.

Thirdly, it highlights the significance of governance boundaries, emphasizing the correlation between party organization governance and governance by non-state shareholders. Third, our findings have important implications for SOE to improve their governance capabilities. In terms of direct effects, our evidence shows that different shareholders or directors from different types of companies can improve the governance level of SOE and be more active in the board resolution process. After introducing the variable of party organization governance, we found that when there are important members of the party organization on the board of directors, the positive impact of non-state shareholder governance on the governance level of SOE is suppressed. If SOEs want to further improve the level of governance, one strategy may be to clarify the boundaries of party organizations’ governance rights, responsibilities, and interests, to make them legal, and to jointly exert the governance effects of mixed ownership reform and party organizations.

Lastly, the study puts forth concrete reform recommendations, offering targeted guidance for mixed ownership reform and promoting the enhancement of governance levels in state-owned enterprises. In summary, this research holds profound significance in advancing theoretical understanding, providing empirical support, and offering specific recommendations for practical implementation.

2. Highlight their economic and research and policy implications. In the discussion of the results please focus on the novel findings and insights vis-à-vis the existing literature.

2. Many thanks to reviewer for his/her attention to our work and constructive advice. This comment is very important for readers to understand our innovation of this article. We emphasized the policy impact of research on the economy. And when discussing the results, the focus was on novel findings and insights relative to existing literature. The details are as follows (The modified content in this section is integrated with the “innovative points” and “practical implications” of the main text):

Economic implications:

Enhancing the Governance Effectiveness of SOEs. We emphasize the positive impact of non-state shareholder governance on the governance level of state-owned enterprises, offering the potential for increased economic vitality and market competitiveness. By reinforcing the role of board resolutions, the governance capacity of non-state shareholders has been proven to enhance the overall governance quality of state-owned enterprises, providing a more effective decision-making mechanism for business development.

Facilitating Mixed Ownership Reform. We underscore the significance of non-state shareholder governance in the context of mixed ownership reform, providing guidance for the introduction of more market-oriented mechanisms and attracting social capital in state-owned enterprises. This is poised to accelerate the progress of mixed ownership reform, encouraging greater involvement of non-state shareholders in corporate governance. Consequently, this momentum propels state-owned enterprises towards a more market-oriented operational framework.

Policy implications:

Strengthening Reform of Corporate Governance Mechanisms. Addressing state-owned enterprises, we advocate for the enhancement of corporate governance mechanisms, ensuring that non-state shareholders or directors possess substantive power and a meaningful voice in corporate decision-making. This serves as guidance for government authorities striving to advance reforms and optimize governance in state-owned enterprises, emphasizing the need to establish governance mechanisms capable of balancing the interests of all stakeholders.

Optimizing the Role of Party Organization Governance. The conclusion highlights the moderating role of party organization governance in board resolutions, offering insights for policymakers. The government should clearly define the role of party organizations in both market and non-market strategic decisions within enterprises. Measures should be implemented to ensure their reasonable contribution throughout the entire governance process, without hindering the participation of non-state shareholders.

New discoveries and insights relative to existing literature:

The study reveals the mediating role of board resolutions in understanding the impact mechanism of non-state shareholder governance on state-owned enterprises. It underscores the critical function of board resolutions in the mediation process. We contend that a more in-depth exploration of the actual decision-making process of the board is essential to draw accurate conclusions from the internal logic and causal relationships. Existing literature predominantly focuses on the direct effects of non-state shareholder governance from the perspectives of equity structure and board composition, lacking a nuanced examination of the micro-level decision-making processes in non-state shareholder governance.

Regulatory Effects of Party Organization Governance. By introducing party organization governance as a boundary condition in the study, it reveals the moderating effects of party organizations on the impact of non-state shareholder governance. This discovery, relatively underexplored in existing literature, provides a new dimension for a deeper understanding of the governance framework within state-owned enterprises.

Response to Reviewer 1

1. The abstract should clearly explain the research motivation, content, methods, conclusions, and significance of the article. Please modify and improve the abstract.

1. We are very grateful to the reviewers for their suggestions and related reviews. Based on the reviewer's suggestions, we have revised and improved the abstract as follows:

With an increasing number of heterogeneous shareholders participating in corporate governance in reality, the assumption of shareholder homogeneity in agency theory is gradually relaxing in the modern field of corporate governance. The policy of mixed ownership reform in China provides empirical evidence for studying heterogeneous shareholder governance. To fully understand the governance effects of non-state shareholders, we employ the ownership proportion held by non-state shareholders among the top ten shareholders and the appointment of directors as measures for non-state shareholder governance. Using a panel fixed-effect model from the perspective of state-owned enterprise (SOEs) party organizations, we examine the impact of non-state shareholder governance on the governance level of state-owned enterprises. The study reveals that non-state shareholder governance positively affects the governance level of SOEs, with board resolutions playing a crucial role in this relationship. However, when important party members serve as directors, the governance effect of non-state shareholders becomes less significant. Based on the above research findings, we recommend further refining corporate governance measures for state-owned enterprises in the context of mixed ownership reform. It is advisable to optimize the governance structure of party organizations, clarify the boundaries between non-state shareholder governance and party organization governance, and advance reforms along the Pareto improvement path. This will contribute to the establishment of a distinctive corporate governance system for state-owned enterprises in China.

2. From the hypothesis, it can be seen that the author divides the governance of non-state-owned shareholders into shareholding structure and board structure, and proposes hypotheses separately from two aspects. However, we do not see the logical differences in the impact of non-state-owned shareholders' shareholding and appointing directors on the corporate governance level in this section, which are explained from reducing agency costs, exerting supervision and balance. It is recommended to explore the impact paths of shareholding structure and board structure on the corporate governance level in a targeted manner. The current analysis is not sufficient.

2. Many thanks to the reviewers for their suggestions. We have added the difference in the impact of non-state-owned shareholder shareholding and the appointment of directors on corporate governance level in the "Theoretical Analysis and Research Hypotheses" section based on the suggestion. Details as follows (The modified content is in the “Theoretical analysis and research hypothesis” section):

Shareholding Structure:

Stakeholder theory suggests that the management activities of a company's business managers balance the interest requirements of each stakeholder in a comprehensive manner (Aguinis and Glavas, 2012). In the internal environment of enterprises, the participation of heterogeneous equity subjects in corporate decision-making can expand the set of opportunities for decision-making. Moreover, it also contributes to the specialization of decision-making and capital and risk-taking, which is conducive to forming a democratic and scientific decision-making mechanism and improving corporate risk control (Wang et al., 2023). As Bennedsen and Wolfenzon (2001) note, when there are multiple heterogeneous shareholders in a company, they can make the right business decisions and strengthen corporate governance capabilities. 

Board Structure:

Based on upper echelons theory, managerial characteristics influence strategic choices of companies (Hambrick and Mason, 1984). Managers with different background traits will have different values and personal perceptions. These factors will directly influence their communication and cooperation at work, indirectly influencing the related decisions and, consequently, the company's behavior. Meanwhile, resource dependence theory notes that board composition is essential to the board's ability to provide governance to the company (Thompson and Alleyne, 2023). As Pfeffer (1972) considers, the composition of the board of directors is not random but a logical response to the company's internal environment.

Aguinis H., Glavas A., “What we know and don’t know about corporate social responsibility: A review and research agenda”. Journal of management, vol.38, no.4, pp. 932-968, 2012.

Wang J, Hu Y, Liao F, Xu S. “Governance of non-state-owned shareholders and corporate capital structure decision: A mechanism test from the opportunistic behavior of management”. PloS one vol. 18, no.1, pp. e0281120, 2023.

Hambrick D.-C., Mason P.-A., “Upper echelons: The organization as a reflection of its top managers”, Academy of management review, vol.9, no.2, pp. 193-206, 1984.

Thompson R M, Alleyne P. Role of a board of directors and corporate governance in a state-owned enterprise”. Corporate Governance: The International Journal of Business in Society, vol.23, no.3, pp. 478-492, 2023.

Pfeffer, J., “Merger as a response to organizational interdependence”, Administrative science quarterly, pp. 382-394, 1972.

3. Why is the sample limited to state-owned manufacturing enterprises in the competitive field? Many studies on non-state-owned shareholder governance are based on a full sample, and the article does not explain the reasons for selecting the sample. The research background discussed in the introduction and the subsequent hypothesis proposed did not focus on specific samples.

3. We are very grateful to the reviewer for reading our manuscript carefully and very useful modification advice. We have added relevant content in the introduction and sample sections respectively. Details as follows:

Introduction

In 2015, the Central Committee of the Communist Party of China (CPC) and the State Council jointly issued the "Guiding Opinions on Deepening the Reform of State-Owned Enterprises." This document outlined a framework for defining functions, categorizing, and progressively implementing reforms in state-owned enterprises (SOEs). The suggested approach involved categorizing SOEs into competitive and non-competitive types, with non-competitive entities further classified as natural monopolies or serving public welfare goals. Non-competitive SOEs primarily align with national policy objectives. Competitive SOEs, on the other hand, prioritize market profitability goals. Considering relevant theories in traditional corporate governance, our research focuses on competitive state-owned enterprises.

The Sample

Taking into account that non-competitive state-owned enterprises are found in industries such as mining, electricity, gas, water production and supply, construction, transportation, warehousing, and postal services, our research primarily focuses on the manufacturing sector.

4. It is recommended to place the empirical model after the "Variable Selection" section, rather than in the empirical results section.

4. We are very grateful for the reviewer's suggestion. Based on the reviewer's suggestion, we placed the empirical model after the variable selection section.

5. There are certain issues with the empirical testing of this article. The "Theoretical analysis and research hypothesis" section elaborates on the impact of shareholding structure and board structure, and suggests testing both aspects simultaneously in the "Baseline regression results" section, rather than focusing solely on shareholding structure.

5. We are very grateful for the questions raised by the reviewer. According to the reviewer's critical suggestions, we included the impact of board structure in the "Baseline regression results" section. The details are as follows：

When the board structure is used as an independent variable, the coefficient pai_nsta is significantly positive. This means that non-state-owned directors can help improve the governance level of state-owned enterprises. Hypothesis H1B is verified.

Table 3: Shareholding structure and corporate governance level

 corporate governance level: gover

 1.OLS 2.RE 3.FE 4.Cluster 1.OLS 2.RE 3.FE 4.Cluster

nsta

 0.274***

(7.41) 0.253***

(6.00) 0.174***

(3.28) 0.253***

(3.14) 

pai_nsta 0.262***

(3.49) 0.241***

(2.72) 0.232**

(2.16) 0.241**

(2.11)

size -0.142***

(-16.83) -0.119***

(-8.66) -0.069***

(-3.39) -0.119***

(-6.03) -0.004

(-1.37) -0.005

(-1.00) 0.004

(0.43) -0.005

(-1.10)

manage 0.129

(1.26) 0.196

(1.55) 0.122

(0.48) 0.196

(1.03) 0.162

(1.60) 0.216*

(1.73) 0.146

(0.58) 0.216

(1.18)

lev 0.0432

(0.70) -0.0748

(-1.04) -0.147*

(-1.79) -0.0748

(-0.79) 0.007

(0.37) 0.019

(0.70) 0.046

(1.30) 0.019

(0.59)

roa -1.542***

(-6.95) -0.665***

(-3.54) -0.487**

(-2.49) -0.665***

(-2.99) -0.123*

(-1.80) -0.166**

(-2.17) -0.149*

(-1.77) -0.166*

(-1.78)

cfo -0.0661

(-0.44) -0.104

(-0.91) -0.0509

(-0.43) -0.104

(-0.77) 0.109**

(2.38) 0.121**

(2.56) 0.125**

(2.48) 0.121*

(1.68)

grow 0.006

(0.49) -0.007

(-0.59) -0.021

(-1.60) -0.007

(-0.55) 0.006

(0.51) -0.007

(-0.59) -0.022

(-1.64) -0.007

(-0.54)

expend -0.0802

(-1.19) -0.117

(-1.52) -0.154*

(-1.82) -0.117

(-1.63) -0.0798

(-1.19) -0.116

(-1.50) -0.154*

(-1.82) -0.116

(-1.60)

_cons 2.725***

(8.15) 2.139***

(6.08) 1.089**

(2.25) 2.139***

(4.10) -0.010

(-0.08) 0.065

(0.46) -0.140

(-0.65) 0.065

(0.60)

year & firm YES YES YES YES YES YES YES YES

N 4101 4101 4101 4101 4101 4101 4101 4101

R2 0.7575 0.6974 0.4909 0.5487 0.5813 0.4778 0.4349 0.4754

Table 4: Instrumental variable approach

 corporate governance level: gover

 1st-stage 2nd-stage 1st-stage 2nd-stage

 nsta gover pai_nsta gover

nsta 0.081***

(4.37) 

meannsta 0.318***

(6.57) 0.069***

(3.98)

 0.147***

(4.20) 

com_port 0.291***

(3.96) 0.121***

(5.37) 

_cons 0.157***

(3.46) 3.710***

(4.35) 0.284***

(5.68) 5.980***

(3.72)

controls YES YES YES YES

year & firm YES YES YES YES

N 4101 4101 4101 4101

R2 0.025 0.473 

First stage F test 157.25*** 158.17*** 

Sargan statistic(p-value) 0.2154 0.2685

6. The limitations of the research and future research directions should be explained in the conclusion of the article.

6. We are very grateful for the questions raised by the reviewer. According to the reviewer's critical suggestions, we supplemented the limitations of our research and future research directions. The details are as follows (The modified content is in the “Conclusions and implications” section)：

This study has several limitations. Firstly, it primarily focuses on the mixed ownership reform of Chinese state-owned enterprises, thus limiting the generalizability of the conclusions to other regions. Secondly, the research relies predominantly on quantitative methods and lacks a diverse methodological approach, such as qualitative studies and case analyses, potentially hindering a comprehensive understanding of the impact mechanisms of non-state shareholder governance. Additionally, the study's timeframe extends only until 2019, failing to capture the dynamic processes of the mixed ownership reform comprehensively and potentially overlooking recent impacts of governance reforms.

Future research can explore several directions: Firstly, there is a need to deepen the study of governance mechanisms, with a focus on understanding the specific impact of non-state shareholders on the operational aspects of governance mechanisms within state-owned enterprises. Secondly, investigating the variations in the governance effectiveness of non-state shareholders across different industries and company sizes would provide valuable insights. Thirdly, conducting more longitudinal studies to track the long-term effects of governance reforms would enhance our understanding of the sustained impact of these reforms. Lastly, extending the scope of research to include other countries and comparing governance models in different nations would offer comparative insights into state-owned enterprise reforms globally. By addressing these research avenues, future studies can achieve a more comprehensive understanding of the impact of mixed ownership reforms on the governance of state-owned enterprises, providing more targeted recommendations and policy support.

Response to Reviewer 2

1. Problem statement of the study must be supported by most recent literature.

1. We are very grateful for the questions raised by the reviewer. According to the reviewer's critical suggestions, we have improved the research question and added support from relevant literature (The modified content is in the “Introduction” section).

The heterogeneous ownership structure has gradually relaxed assumptions in existing corporate governance policies and practices based on traditional theories. With different resource endowments and governance objectives, it ultimately affects a firm's financial behavior and operational performance (Aghion et al., 2013). Governance by non-state shareholders falls under the category of heterogeneous shareholder governance, and currently, there is limited theoretical research in the field of corporate governance regarding heterogeneous shareholder governance behavior (Connelly et al., 2016). For instance, Williams and Ryan (2007) found that greater disagreements among heterogeneous shareholders may pose challenges to executive performance evaluation and monitoring. Additionally, the presence of heterogeneous shareholders makes it difficult to balance the diverse interests of company shareholders, leading management to prioritize the interests of shareholders whose preferences and risk tolerances align more closely with their own. Similarly, Goranova and Ryan (2022) emphasize that executives' vested interests do not necessarily aggregate and balance the interests of heterogeneous shareholders but rather prioritize the interests of those shareholders whose interests align more closely with their own. Therefore, addressing issues related to heterogeneous shareholder governance is of utmost importance.

Connelly B.-L., Haynes K.-T., Tihanyi L., et al., “Minding the gap: Antecedents and consequences of top management-to-worker pay dispersion”. Journal of Management, vol.42, no.4, pp.862-885, 2016.

Williams C.-C, Ryan L.-V., “Courting shareholders: The ethical implications of altering corporate ownership structures.” Business Ethics Quarterly, vol.17,no.4, pp.669-688,2007.

Goranova M, Ryan L.-V., “The corporate objective revisited: the shareholder perspective.” Journal of Management Studies, vol.59, no.2, pp.526-554, 2022.

2. Significance of study must be specified in introduction section.

2. We are very grateful for the questions raised by the reviewer. According to the reviewer's critical suggestions, we have improved the Significance of study in introduction section. Specifically, as follows (The modified content is integrated into the innovation points):

Firstly, the study deepens the understanding of mixed ownership reform, revealing its mechanisms and effects in enhancing the governance efficiency of SOEs. We add to the literature documenting the economic benefits of non-state shareholder governance in SOEs. Several recent studies have examined the impacts of non-state shareholder governance on innovative decision-making (Wang et al., 2021), operating performance (Guan et al., 2021), and technology innovation (Zhang et al., 2020). Research on non-state shareholder governance and governance level of SOE, what is not been thoroughly examined in detail thus far are the reasons for such outcomes. In addition, although there are many literatures on the level of corporate governance, most of them study corporate governance as a pre-variable (Ammann et al., 2011; Fu and Hong, 2018; Chu et al., 2019). Taking Zhejiang enterprises as the research object, Du et al. (2021) analyzed the governance effect evaluation of the mixed ownership reform, but there is insufficient research on the improvement path of its governance capacity. We take corporate governance level as a post-variable It is very important in SOE to explore the impact of non-state shareholder governance on governance level. We complement these studies by providing evidence that non-state shareholder governance promotes governance level of SOE.

Further, we identify the mechanisms of non-state shareholders on governance level from the perspective of board resolutions. One of the most critical aspects of our results is that, in addition to highlighting the influence of non-state shareholder governance on governance level, we also represent indirect evidence of the influence of non-state shareholder governance and board resolutions on governance level of SOE. Thus, our results highlight how non-state shareholder governance interacts with board resolutions and the governance level.

Secondly, by means of policy measures, the study provides empirical support for heterogeneous shareholder governance, bridging the gap between theory and practice. We place party organization governance within the framework of non-state shareholder governance and SOE governance. Party organization and mixed ownership reform are unique systems of Chinese SOE, and most of the relevant literature studies them separately. As important stakeholders, whether important members of the party organization and non-state shareholders/directors can exert the governance effect of "1+1>2" remains to be tested. Therefore, combined with the mixed ownership reform system, we incorporate the party organization into the analytical framework of corporate governance at the micro level of SOE, explore the relationship between non-state shareholders and party organization governance, and provide theoretical and empirical support for it.

Thirdly, it highlights the significance of governance boundaries, emphasizing the correlation between party organization governance and governance by non-state shareholders. In terms of direct effects, our evidence shows that different shareholders or directors from different types of companies can improve the governance level of SOE and be more active in the board resolution process. After introducing the variable of party organization governance, we found that when there are important members of the party organization on the board of directors, the positive impact of non-state shareholder governance on the governance level of SOE is suppressed. If SOEs want to further improve the level of governance, one strategy may be to clarify the boundaries of party organizations’ governance rights, responsibilities, and interests, to make them legal, and to jointly exert the governance effects of mixed ownership reform and party organizations.

Lastly, the study puts forth concrete reform recommendations, offering targeted guidance for mixed ownership reform and promoting the enhancement of governance levels in state-owned enterprises. In summary, this research holds profound significance in advancing theoretical understanding, providing empirical support, and offering specific recommendations for practical implementation.

3. Most recent citations must be provided in literature section. It was difficult to find any citation from 2022 and 2023, authors are suggested to add literature from 2022 and 2023 to support research objectives and research hypothesis.

3. We are very grateful for the questions raised by the reviewer. According to the reviewer's critical suggestions, we have added relevant literature to support the research objectives and hypotheses:

Recent years witnessed a surge in heterogeneous shareholder governance of firms on a world-wide scale. Goranova and Ryan (2022) argues that participation of heterogeneous shareholders is a preferred institutional model for the development of firms towards good discovery. Under China's SOE reform system, more and more heterogeneous shareholders are involved in the management and governance of SOEs (Ma and Huang, 2023; Qiao et al., 2023; He et al., 2023). Studies find that firms supervised or managed by shareholders or boards with heterogeneous backgrounds tend to the good governance characteristics (Wang et al., 2023). They are fewer agency costs (Wang et al., 2021), are more technological innovations (Xu et al., 2023), and are more efficient with investments (Fan et al., 2022). Among the reasons mentioned is that heterogeneous shareholders or directors improve corporate governance and management practice. This phenomenon is because the heterogeneous shareholder or director mechanism has the characteristics of flexibility, independent decision-making, market sensitivity, cost control, and perfect management. Building on this evidence, we examine the impact of non-state shareholder governance on the governance level of SOEs.

Ma X.-X., Huang X.-S., “Mixed ownership reform of state-owned enterprises and dual-sided optimization of labor investment efficiency”. Finance and Trade Research, vol.34, no.11, pp.84-98, 2023.

Qiao C.-X., Ma Y.S., Liu Y.-Z., “Governance of non-state shareholder s and innovation of state-owned enterprises: an inverted U-shaped relationship and its formation mechanism test”. Reform, no.2, pp.118-138, 2023.

He Y., Yang L., Wen W., “Can the participation of non-state-owned shareholders in governance improve the “market rationality” of financing behavior of state-owned enterprises: evidence from dynamic adjustment of capital structure”. Nankai Business Review, vol.26, no.1, pp.118-133+158+134-135, 2023.

Wang J, Hu Y, Liao F, Xu S. “Governance of non-state-owned shareholders and corporate capital structure decision: A mechanism test from the opportunistic behavior of management”. PloS one vol. 18, no.1, pp. e0281120, 2023.

Wang, H., Wang W., Alhaleh S., “Mixed ownership and financial investment: Evidence from Chinese state-owned enterprises”, Economic Analysis and Policy, 70:159-171. vol.70, pp. 159-171, 2021.

Xu D.-D., Li X.-L., Wang J., Can the governance of non-state-owned shareholders promote the green technology innovation of state-owned enterprises—— Empirical research based on mixed-ownership reform. Business Review, vol.35 no.09, pp.102-115, 2023.

Fan, R., J. Pan, M. Yu, and H. Gao. “Corporate governance of controlling shareholders and labor employment decisions: Evidence from a parent board reform in China”, Economic Modelling, 105753. 2022.

4. A short review of governance policies implemented in China must be explained to provide readers an insight about it.

4. We are very grateful for the questions raised by the reviewer. According to the reviewer's critical suggestions, We reviewed relevant policies in China in introduction section: 

In 2013, the Third Plenary Session of the Eighteenth Central Committee proposed actively promoting the integration of heterogeneous capital, strengthening collaborative cooperation, and further emphasizing the importance of mixed ownership. This led to the development of more enterprises as mixed ownership entities with the convergence of heterogeneous capital. As the top-level document for state-owned enterprise reform, in 2015, specific reform measures were put forward by the Central Committee of the Communist Party of China and the State Council regarding how state-owned enterprises should undergo mixed ownership reform ("Guiding Opinions on Deepening the Reform of State-Owned Enterprises"). This provided favorable conditions for the practical implementation of mixed ownership reform. In October 2016, General Secretary Xi Jinping emphasized at the National Party Building Work Conference for State-Owned Enterprises, "Adhering to the Party's leadership over state-owned enterprises is a major political principle that must be consistently followed. Establishing a modern enterprise system is the direction of state-owned enterprise reform and must also be consistently followed." He specifically highlighted the fundamental principle of "integrating the Party's leadership into every aspect of corporate governance, embedding the enterprise party organization into the corporate governance structure." After years of state-owned enterprise reform, in 2023, the Central Committee of the Communist Party of China issued the "Comprehensive Implementation of State-Owned Enterprise Reform and Deepening Enhancement Action," emphasizing the importance of "the significant discourse on the reform and development of state-owned enterprises and Party building as a primary task." It is evident that governance by non-state shareholders and party organization governance remains a focal issue in the current field of corporate governance in China. Simultaneously, guided by stakeholder theory, China's state-owned enterprise reform policies provide a valuable perspective for the traditional corporate governance domain.

5. All variables of the study should be further explained in a table to make it more convenient for readers to understand study variables.

5. We are very grateful for the questions raised by the reviewer. According to the reviewer's critical suggestions, we further explained in a table to make it more convenient for readers to understand study variables.

Table 1: Variable definitions

Variable Symbol Definition

Governance level of SOE gover Based on the research of Zhou et al. (2020), we use the principal component analysis method to construct comprehensive indicators to measure the governance level of state-owned enterprises from three aspects (7 variables): supervision (proportion of independent directors, size of the board of directors, institutional shareholding ratio and equity balance), incentive (proportion of executive compensation and executive shareholding), and decision-making (whether the chairman and the general manager have two functions in one)

Non-state shareholder governance nsta Ratio of non-state shareholders among the top ten shareholders

 pai_nsta Natural logarithm of one plus the number of board members appointed by non-state shareholders

Board resolution behavior dissent If the SOE has board objections in the annual board resolution announcement, it will be 1, otherwise it will be 0

Party organization governance party If the secretary or deputy secretary of the party committee serves as a member of the board of directors (chairman, vice chairman or director), it is defined as 1; otherwise, it is 0.

firm size size Natural logarithm of the book value of total assets at the end of the year

management's shareholding manage Number of shares held by management/total share capital of the company

assets and liabilities lev The ratio of total liabilities to total assets

return on assets ratio roa Ratio of net cash flows from operating activities to total assets

net cash flow cfo Net cash flow from operating activities/total assets

business growth grow operating income growth rate

capital expenditures expend Capital Expenditure/Total Assets

6. Sampling technique for sample selection is missing in research methodology section. Provide sampling technique with justification.

6. We are very grateful for the questions raised by the reviewer. According to the reviewer's critical suggestions, we have provided a reasonable sampling technique, the PSM method, in the "robustness testing" section. Specifically, as follows:

As a robustness check, we re-run our model a propensity score matched sample to mitigate this concern. Specifically, we use logit model to regress our indicator variable board governance (pai_nsta) on control variables in model (1) and estimate the propensity score that a SOE have a board member with non-state background. Secondly, we match each treatment SOE ( non-state background =1) with a control SOE ( non-state background =0) with the closest propensity score. 

7. Data for study is collected from 2009 to 2019. It is suggested to increase the sample time period upto 2022 to make the study more authentic.

7. We sincerely appreciate the questions raised by the reviewers. The suggestions provided by the reviewers are highly practical for our research. Although we have commenced the process of compiling and summarizing relevant data from 2019 to 2023, the collection and integration of data on non-state shareholders from company annual reports, particularly their proportions among the top ten shareholders and the appointment of non-state shareholder-appointed directors, is a labor-intensive task that cannot be completed in the short term. We acknowledge the challenge of updating our data, as indicated in the "Limitations and Future Research Directions" section, and we are committed to addressing this issue in our future research. Once again, we express our gratitude to the reviewers for their valuable suggestions that will undoubtedly enhance the quality of our study. 

8. Statistical techniques used in the study must be explained in research methodology section along with all required quantitative models or equations.

8. We are very grateful for the questions raised by the reviewer. According to the reviewer's critical suggestions, we have provided an explanation:

Using a panel fixed effects model. ...Additionally, we control firm fixed effects, year fixed effects, The coefficient of interest is α1. We expect the coefficient to be positive, implying that non-state shareholder governance can help improve the governance level of SOEs.

9. All results in data analysis section must be supported by similar studies carried out in China or in the Asian region.

9. We are very grateful for the questions raised by the reviewer. According to the reviewer's critical suggestions, we have added relevant literature to support our research based on all the results of data analysis.

The descriptive statistics of the remaining control variables are generally consistent with existing studies (Zhou et al., 2020; Qin and Fuan, 2021).

Qin H.-L., Fuan S.-C., “Mixed reform of state-owned enterprises, governance structure and cash dividends - from the perspective of corporate governance”. Review of Investment Studies, vol.40, no.11, pp. 37-58, 2021.

---

## [Decision Letter · Decision Letter 1]

24 Mar 2024

The Influence of Non-state-owned Shareholder Governance on the Governance Level of State-owned Enterprises——Based on the Perspective of Board Resolution Behavior and Party Organization Governance

PONE-D-23-33605R1

Dear Dr. Zhang,

We’re pleased to inform you that your manuscript has been judged scientifically suitable for publication and will be formally accepted for publication once it meets all outstanding technical requirements.

Kind regards,

Wajid Khan

Academic Editor

PLOS ONE

Additional Editor Comments (optional):

Reviewers' comments:

Reviewer's Responses to Questions

**Comments to the Author**

1. If the authors have adequately addressed your comments raised in a previous round of review and you feel that this manuscript is now acceptable for publication, you may indicate that here to bypass the “Comments to the Author” section, enter your conflict of interest statement in the “Confidential to Editor” section, and submit your "Accept" recommendation.

Reviewer #1: All comments have been addressed

Reviewer #2: All comments have been addressed

2. Is the manuscript technically sound, and do the data support the conclusions?

Reviewer #1: Yes

Reviewer #2: Yes

3. Has the statistical analysis been performed appropriately and rigorously? 

Reviewer #1: Yes

Reviewer #2: Yes

4. Have the authors made all data underlying the findings in their manuscript fully available?

Reviewer #1: (No Response)

Reviewer #2: Yes

5. Is the manuscript presented in an intelligible fashion and written in standard English?

Reviewer #1: Yes

Reviewer #2: Yes

6. Review Comments to the Author

Reviewer #1: After the author's modifications, the previous issue has been greatly improved. Here is a final suggestion: it would be better if the adjusted R2 could be presented in the regression results.

Reviewer #2: (No Response)

7. PLOS authors have the option to publish the peer review history of their article (what does this mean?). If published, this will include your full peer review and any attached files.

Reviewer #1: No

Reviewer #2: No

---

## [Editor Report · Acceptance letter]

27 Mar 2024

PONE-D-23-33605R1 

PLOS ONE

Dear Dr. Zhang, 

I'm pleased to inform you that your manuscript has been deemed suitable for publication in PLOS ONE. Congratulations! Your manuscript is now being handed over to our production team.

Kind regards, 

on behalf of

Dr. Wajid Khan 

Academic Editor

PLOS ONE